# Beamforming Optimization in Internet of Things Applications Using Robust Swarm Algorithm in Conjunction with Connectable and Collaborative Sensors

**DOI:** 10.3390/s20072048

**Published:** 2020-04-06

**Authors:** Mohammed Zaki Hasan, Hussain Al-Rizzo

**Affiliations:** 1College of Computer Science and Mathematics, University of Mosul, Mosul 41002, Iraq; 2Systems Engineering Department, Donaghey College of Engineering & Information Technology, University of Arkansas, Little Rock, AR 72701, USA; hmalrizzo@ualr.edu

**Keywords:** robust optimization, internet of things, collaborative sensor, beamforming, antenna array

## Abstract

The integration of the Internet of Things (IoT) with Wireless Sensor Networks (WSNs) typically involves multihop relaying combined with sophisticated signal processing to serve as an information provider for several applications such as smart grids, industrial, and search-and-rescue operations. These applications entail deploying many sensors in environments that are often random which motivated the study of beamforming using random geometric topologies. This paper introduces a new algorithm for the synthesis of several geometries of Collaborative Beamforming (CB) of virtual sensor antenna arrays with maximum mainlobe and minimum sidelobe levels (SLL) as well as null control using Canonical Swarm Optimization (CPSO) algorithm. The optimal beampattern is achieved by optimizing the current excitation weights for uniform and non-uniform interelement spacings based on the network connectivity of the virtual antenna arrays using a node selection scheme. As compared to conventional beamforming, convex optimization, Genetic Algorithm (GA), and Particle Swarm Optimization (PSO), the proposed CPSO achieves significant reduction in SLL, control of nulls, and increased gain in mainlobe directed towards the desired base station when the node selection technique is implemented with CB.

## 1. Introduction

Beamforming techniques are gaining an increasing research interest in recent years due to the growth of Internet of Things (IoT) applications that require establishing connections for long distances among sensors and a remote base station [1]. The popularity of these applications stems from the demand of increasing throughput accompanied by decreasing size of electronic devices [2]. This requires that all objects or devices in the IoT to be equipped with directional antennas which form a virtual antenna array.

Due to a variety of reasons including their small size, low cost, ease of deployment (deterministic or random) and simplified transmission-related protocols in terms of medium access channels and routing [3], it has been found that the antenna array is a key technology for a broad spectrum of future IoT applications ranging from weather forecasting or complex industrial plant monitoring to military surveillance [4].

The next generation of Wireless Sensor Networks (WSNs) target the design of sensors systems that satisfy the requirements of IoT applications taking into consideration recent advancement of beamforming technology in 5G wireless systems, whereby the omnidirectional antenna may either be replaced by directional ones or can work in tandem with the same motes [5].

Although omnidirectional antennas lead to poor network performance due to interference and short transmission range, previous studies investigated network connectivity using omnidirectional antennas [6]. However, directional antennas have become mandatory in future generations of WSNs requiring high network connectivity [7]. Employing a directional antenna model may be very complicated when optimal bounds of network connectivity are sought for specific applications. Such applications may require low-power sensors with simple hardware which are randomly deployed over an area of interest. Furthermore, several IoT applications are required to transmit the acquired data over long distance using multihop transmission resources available only at sensors. Therefore, this can be costly for battery-powered sensors. Hence, one major problem is to obtain optimal bounds of network connectivity between battery-powered sensors and the intended base station which might be allocated far away from the transmission range of sensors. In other words, directional antennas are sensitive in terms of directional alignment and thus every sensor must steer its mainbeam to each other for a successful connectivity [8].

There are several challenges that must be considered when a model is developed in the realm of virtual beamforming in IoT environments. For example, each sensor must connect with others as well as with the wireless devices involved in the IoT environment that may use different technologies for communicating with sensors. Generally, coexistence is a severe challenge in the IoT model, since the devices that are trying to communicate may interfere with each other resulting in data loss, rendering the sensors useless. Meanwhile, there is a substantial work on coexistence for collaborative beamforming; the forecast includes thousands of devices in a narrow range. These applications in IoT also impose challenging requirements on manufacturing as far as creating large-scale deployments is concerned.

To overcome this problem, Collaborative Beamforming (CB) can be used for reliable communication over long multihop distance. CB is considered to be a branch of cooperative communication where randomly located sensors cooperate with each other to form a virtual antenna array. This collaboration is presented as a set of sensors operating in half-duplex mode forming an antenna array where their input signals are multiplied with properly selected complex beamforming weights. CB extends the transmission range of collaborative sensors by using sensors which jointly transmit to generate a coherent directive beam for transmission to a given base station.

In the context of our paper, sensors within IoT environment are considered to be an area of interest that acts as a collaboratively distributed antenna array that forms a specific network topology such as linear, mesh, ring, tree or even random according to the connectivity among the collaborative sensors. These collaborative sensors adjust the initial phases of their carriers such that signals from individual sensors add constructively and form a beam toward the direction of the intended base station. Moreover, CB can increase the coverage area of WSN, and can be also viewed as an alternative scheme to the multihop relay communications. Consequently, CB brings many advantages compared to the multihop relay communications for several reasons. First, no dependency of communication quality on individual sensors. Secondly, CB enables operating a single-hop directly to the base station, thus, it reduces the data overhead and delay. Finally, CB in directional antenna transmission achieves higher connectivity compared omnidirectional transmission with the same transmit power. Beside these challenges and advantages, the one concern in CB design is the uncontrolled sidelobes of the beamforming pattern due the following reasons:Random deployment whereas the sensors are randomly distributed over a specific area which collaboratively form an antenna array for beamforming purposes which is referred to as virtual random antenna array [4]. The location of each sensor must be considered on its own. Moreover, these randomly located sensors must be controlled by a cluster head. Hence, the locations, phase offset and transmit capabilities of each sensor must be known quantities to be taken into consideration during weight calculations,Synchronizing the distributed sensors to achieve the desired beamforming, andHardware limitations and processing of sensors which are restricted to simple circuitry.

For these reasons, it might be concluded that the method for determining the complex weights should be carried out individually by each sensor without sharing information. Meanwhile, if the sensors have abilities to share and exchange information, initial phases can be used to calculate the weights based on global information in terms of the positions and disseminated through the network by cluster heads. This method strategically controls the deployment of distributed sensors according to pre-calculated and optimized global parameters of each sensor using various metaheuristic optimization algorithms.

This paper proposes a local neighbor search method based on robust Canonical Swarm Optimization (CPSO) algorithm to investigate the effects of random deployment of disjointed sensors to achieve beamforming with sharp beamwidth while minimizing interference from undesired interferers. The resultant beampattern is crucial to decide whether the attainment of a CB technique is considered feasible with respect to the requirements of IoT applications. We claim that the random deployment and sharing of information among the selected participating sensors motivate CB to determine the characteristics of the beampattern based on the requirements of IoT applications. Furthermore, various network topologies of participating sensors produce different effects on transmission gain. The algorithm proposed in this paper creates a network topology of a virtual antenna array whereas the weight coefficient of each sensor is adjusted to control the reduction of sidelobe levels (SLL) while satisfying the connectivity among them that is constrained by the distance among the collaborating sensors. We establish the CPSO algorithm using directional antenna model [6] to analyze local connectivity i.e., the probability of isolation of the sensor, as well as the overall global connectivity by evaluating the probability of connectivity of at least one connected path that identifies the network topology in terms of ring, tree, mesh, and random from the entire network. The main contributions of this paper are summarized as follows:A metaheuristics algorithm is developed in terms of canonical swarm to investigate the effects of random deployment of disjointed sensors to solve the SLL reduction and maximum mainlobe in CB.A node selection scheme is formulated based on CPSO to optimize beampattern of the array formed by collaborative sensors by carrying out an exhaustive search over all possible network connectivities of sensors in the network.Compare the beampattern characteristics in the case of uniform linear, mesh and non-uniform array sensor distributions taking into consideration the network connectivity.

The rest of the paper is organized as follows. Section 2 provides a literature review of existing techniques for CB in sensor networks. Section 3 presents the system model for the antenna array. Section 4 and Section 5 introduce the channel and connectively models of the proposed algorithm, respectively. Meanwhile, Section 6 presents sidelobe reduction control through the node selection technique. The simulation results are introduced in Section 7 and compared against existing techniques in terms of conventional beamforming, including convex optimization, Genetic Algorithm (GA), and Particle Swarm Optimization (PSO). Finally, Section 8 concludes the paper.

## 2. Related Works

Several algorithms were proposed to control the sidelobes of antenna arrays whereas the information for each sensor is known. Initially, these works focused on different approaches of sharing information of the same problem. In [9,10,11] the statistical analysis of the beamforming characteristics is reported using various deployments of sensors. The authors in [9] derived the average properties of beamforming for uniformly distributed arrays sharing phase information. Meanwhile the authors in [11] showed that Gaussian distributed sensors provide better average of beampattern properties compared to uniformly deployed sensors. Consequently, in [10] controlling and reducing the sidelobe level was proposed using random node selection algorithm with low-rate feedback. The authors in [12] show that distributed CB can be used in WSN to perform energy-efficient communications.

There are several studies on Ad-hoc/sensor network connectivity [8,13,14], and cognitive radio network [15,16,17], using omnidirectional and directional antenna models. More specifically, the authors in [18] reveal the importance of selecting the directional antenna model using a simple channel model which is tractable in the performance analysis. The authors in [19] provided statistical analysis of the beamforming characteristics through a node selection algorithm of group of sensors that forms a ring network topology. It should be noted that when using the directional antenna model within WSN, it becomes difficult for each sensor to share the global information with other neighbors. Therefore, in [20] the authors developed a fully distributed directional antenna array discovery mechanism to find collaborative sensors. A discrete-time Markov chain is used for analyzing the performance of the antenna array to direct and indirect neighbor discovery as well as selection of feedback policies jointly which helps to decrease collisions and reduce the latency in the network. On the other hand, when a sensor has a fixed topology within a cluster, it may be reasonable to assume that these sensors are connected. However, this is not a well-justified assumption to be used in optimizing the CB performance since it entails frequent and dynamic changing topology of connectable and collaborative sensors. For instance, it has been shown that if CB is considered without optimization, the sensor network may be using more energy than strictly required. Hence, convex optimization has been used to solve the beampattern optimization problem [21].

Research on beamforming employing optimization in connectable and collaborative sensors predominantly undertook either weight coefficient perturbation or the deployment of node perturbation with different objective functions in terms of nulling, minimizing SLLs and maximize the directivity. The authors in [22] proposed the use of cross-entropy optimization for optimally selecting the collaborative sensors to reduce the sidelobe level with minimal computational complexity. An effective robust adaptive beamforming method is presented in [23]. The authors constructed a tapered covariance matrix to broaden the width of nulls for interference signal sources. Moreover, the multiple quadratics inequality constraints outside the search space of proposed solution of beamforming are used to reduce the sidelobe level below a prescribed threshold value. Meanwhile, the $l_{0}$-norm constrained normalized least-mean-square (CNLMS) adaptive beamforming algorithm is proposed in [24] to control the sparsity of the antenna array. The authors formulated the beamforming optimization problem using the $l_{0}$-norm penalty as constraint to force the quantities of multiple antennas to a certain number in order to control the sparsity by selecting a suitable parameter of the convergence speed of the algorithm. The authors in [25] improved the null broadening adaptive beamforming algorithm by realization of the reconstruction of interference-pulse-noise covariance matrix. The proposed algorithm is based on covariance matrix reconstruction method (ABA-CMR) to insert some virtual interference around the incident angle of inference in order to get better performance.

Several research work have been reported based on metaheuristic algorithms such as genetic algorithm [26], particle swarm optimization (PSO) [27], and other metaheuristic algorithms [28] adopting the node selection method on various space topologies of antennas array such as uniform circular [29], linear or mesh array [30], and random array [31] for reducing the sidelobes, nulls control, and minimizing beam width of the beampatten. Specifically, the PSO algorithm is inspired by social behavior patterns of organisms that live and interact within large groups. It could be easily implemented and applied to solve various optimization problems or problems that can be transformed to function optimization. The main strength of PSO is fast convergence, which compares favorably with many other global optimization algorithms such as CPSO, fully informed particle swarm optimization, and multi-swarm optimization algorithms [32,33,34,35,36]. However, CPSO provides a formal proof that each particle converges to a stable point using stochastic process theory. Hence, CPSO can analyze the stochastic convergence of the particle swarm and corresponding parameter selection guidelines. It has been shown that the trajectories of the particles oscillate as different sinusoidal waves and converge quickly [37,38].

The authors in [21] presented a solution for the single look direction of the antenna array problem using an interior-point method to define the connections with specific a geometry. All large-scale interior-point implementations use direct decomposition to solve the reduced Newton system. This can be done by the symmetric factorization of a quasi-definite system. Once the problem is defined as a convex optimization problem, it can be solved in a relatively short time and less space than most randomized incremental algorithm using well-known efficient convex solvers. However, one drawback of direct factorization solvers is that in some situations a sufficient/desired numerical accuracy cannot be achieved. Because the convex optimization algorithm is a local minimizer solver of a convex objective function over a convex set, it needs gradient information. Similar to most of traditional optimizers, the algorithm is unfit for complex multimodal problems and non-differentiable optimization problems. For example, in various IoT applications, it may be very difficult to control the parameters with an arbitrary precision.

Generally, the complex multimodal problems can be modeled as a linear convex optimization problem to obtain optimal and accurate solution. It is not easy to determine whether a function is convex or not. Thus, the richness of convex functions for solving these problems is demonstrated by the connection between the convexity set and the objective function from classical analysis. This connection is given by several theorems to guarantee that the convexity is the main step in the optimization method [39]. Moreover, the choice of optimization methodology depends on several limitations imposed by the sensor network such as the number of sensors deployed in a specific area and the application scenarios in IoT. Existing algorithms for node selection may cause some sensors in the network to be used more often than others and hence exhaust their batteries. If all sensors selected from an excessively large area in the network are used to generate a narrow mainlobe, then the lifetime of network might exhaust as well as the lifetime of frequently used sensors. The main costs for these algorithms are computing distances. To isolate the effect of randomization on time efficiency, we propose changing the process of randomization of swarm performance beyond convex optimization by selecting the further points i.e., solutions from an outside research space. Whether each selected solution swapped with revocable solution, then it could create many other new solutions for multimodal problems. Therefore, the algorithm proposed in this paper is based on minimization of sidelobes and maximization of mainlobe via coefficient weight perturbation with the ability of randomly selecting the sensors while providing improvement in connectivity for sensors deployed over a specific cluster area.

## 3. System Model

In conventional beamforming, the elements of an antenna array are arranged in a predefined regular configuration such as linear or circular. Meanwhile, in CB where each cluster contains randomly distributed sensors acting as a virtual antenna array in an Ad-hoc manner with no common controller, i.e., the weights are synchronized at each sensor. Then, the whole clusters over a distributed area simultaneously transmit a weighted signal which constructively combines in the direction of the intended base station.

We consider a system model where a group of randomly deployed sensors are co-located in the (x,y) horizontal plane. Each sensor has a transmission radius $r_{o}$ as shown in Figure 1 which illustrates the cluster head, collaborating sensors and intended base station. All sensors jointly form a topology of such a network which is dynamically changing in an IoT environment due to the signal variations. However, from the topology point of view, a WSN can be represented as a graph G(V,E), where the sensors correspond to the vertices and the paths between theses sensors correspond to the edges of the graph. Table 1 lists the symbols used throughout the paper along with their definitions.

The number of direct neighbors of a given cluster head is defined as the degree of the connectivity. The simplest case is taken into consideration as an initial step in which it is assumed that the graph consists of a large number of scattered sensors in the coverage area and paired up whenever calculating the probability that a pair of sensors (ı,ȷ)∈κ separated by Euclidean distance $d_{ıȷ}$ is connected, where $d_{ıȷ}$ is given by
(1)$$d_{ıȷ}=\sqrt{{\left(x_{ı}-x_{ȷ}\right)}^{2}+{\left(y_{ȷ}-y_{ȷ}\right)}^{2}}$$
where $x_{ı}$, $y_{ȷ}$ are the coordinates of *k* connecting sensors. The hop-distance between two connected sensors can be calculated as [37]
(2)$$r_{o}=\sqrt{d_{ıȷ}\frac{3\varepsilon f_{s}}{2E_{elec}}}$$
where $E_{elec}$ is the energy dissipation rate to run the radio, $\varepsilon f_{s}$ is the one-path model for the transmitter amplifier. Meanwhile $\varepsilon_{mp}$ is the multipath model for the transmitter amplifier. In other words, any two sensors can directly communicate with each other if and only if their Euclidean distance is smaller than a given threshold $r_{o}$. Suppose a graph G is said it to be connected if and only if at least a path exists $\left(x_{ı\kappa },y_{ȷ\kappa }\right)$ between any pair of sensors *ı* and *ȷ*, where ${\left\{ı,\ldots ,ȷ\right\}}_{ı\neq ȷ\in ℵ\subseteq N}$. When there is no path between any pair of the sensors, the network is said to be disconnected.

We express the sensor coordinates as $(x_{ı},y_{ȷ}),ı,ȷ=1,\ldots ,N$, where ı≠ȷ and κ∈ℵ⊆N. The sensors are chosen as linear, mesh, and randomly distributed in a circle with radius *R* and the spacing between sensors is λ/2 [40]. Meanwhile, the random deployment is assumed to be Gaussian distributed with zero mean and variance $\sigma_{o}^{2}$. Initially, one of the collaborative sensors is selected to act as a cluster head, which then serves as the geometrical reference point for all collaborative sensors in the respective cluster. The cluster head broadcasts Hello messages to all its neighbors. Each connected sensor receives the message, and align its signal phase with reference to the cluster head by multiplying the message by a complex weight. This process is performed during information sharing among the collaborative sensors to ensure that they have aligned the signal to be added constructively at the direction of base station. At the end of this stage, the CB stage starts by the source sensor and its connected collaborators which simultaneously transmit the data packets with a field amplitude of $\xi_{\kappa }$. Due to the phase alignment among sensors during the sharing phase, the data packets add constructively at the intended location.

The channel characteristics in various environments such as outdoor, or outdoor-to-indoor IoT applications, i.e., scenarios such as open-loop or closed-loop in which CB is needed are exposed to large-scale fading which is considered to be a dominant factor for the channel among these collaborative sensors and base station. Therefore, the channel coefficient for each κth collaborative sensor which servers as connectable source–destination pair is multiplied by the corresponding weight to align the phase of the signal. This ensures that signals from all collaborative sensors are in phase toward the direction of the intended base station. Moreover, a closed-loop scenario is considered where the phase alignment is done by compensating the distance between the collaborative sensors and the intended base station, with respect to the cluster head.

Assume the location of the intended location is expressed in spherical coordinate system $\left(r_{o},\theta_{o},\phi_{o}\right)$, where the elevation and azimuth angles correspond to θ∈[0,π] and ϕ∈[−π,π], respectively. The corresponding coordinates expressed as $r_{\kappa }=\sqrt{x_{\kappa }^{2}+y_{\kappa }^{2}}$, and $\psi_{\kappa }=\tan^{-1}\left(\frac{y_{\kappa }}{x_{\kappa }}\right)$ possess Rayleigh and uniform distributions, respectively. Without any loss of generality, we set $\phi_{0}=90$, therefore the Euclidean distance between the κth sensor and a point located in (x,y) plane at coordinates $\left(A,\phi_{0}\right)$ is defined as $d_{\kappa }≜\sqrt{A^{2}+r_{\kappa }^{2}-2r_{\kappa }A\cos\left(\phi -\psi_{\kappa }\right)}∼A-r_{\kappa }\cos\left(\phi -\psi_{\kappa }\right)$, where $A\gg r_{\kappa }$ in the far-field region. The array factor for *N* sensors is expressed as [18]
(3)$$AF^{\kappa }\left(\phi /r,\theta \right)≜\sum_{\kappa =1}^{N}\exp^{ȷ\varphi_{\kappa }}\exp^{-ȷ\frac{2\pi }{\lambda }d_{\kappa }\left(\phi \right)}$$
whereas λ denotes the wavelength and $\varphi_{\kappa }$ is the initial phase of the κth sensor carrier frequency. The phase delay is defined as $\theta_{r}=\frac{2\pi }{\lambda }d_{\kappa }\left(\phi \right)$ at point $\left(A,\phi_{0}\right)$. The gain of an antenna can be expressed in a spherical coordinate system as follows [15]
(4)$$G\left(\theta ,\phi \right)=\eta \frac{u^{\kappa }\left(\phi /r,\theta \right)}{u_{\left(o\right)}^{\kappa }}$$
where u(θ,ϕ) is the radiation intensity in a given direction (θ,ϕ), and η is the efficiency factor, which is set to be one since each antenna is assumed to be lossless and $u_{\left(o\right)}^{\kappa }$ is the radiation intensity of an isotropic radiator. The isotropic antenna model has been frequently used in WSN to model several IoT applications [18,37,41]. Figure 2 illustrated the deployment of the elements along a line, rectangle (mesh), or random with the corresponding distance between two neighboring connecting elements that achieves both connectivity and full coverage.

Applying isotropic antennas ensures the connectivity for example in Figure 3 if sensor (A) connects to sensor (B), then sensor (B) must connect to sensor (A). Nevertheless, it is not guaranteed for directional antennas. Sensor (A) might point to a sensor (B), but the (B) might connect elsewhere such as sensor (D). Sensor (C) has a packet for sensor (A), but the (C) can not transmit the data, since sensor (A) is connected to sensor (B). Therefore, it seems that the connectivity of the network requires global knowledge of all sensors’ directions for which several challenges arise. The first challenge is interference caused by higher gain. Meanwhile, the second is the optimal deployment rules that apply to every sensor to ensure the connectivity with as few other sensors as possible. We consider three kinds of antenna arrays, Uniform Linear Array (ULA), Uniform Mesh Array (UMA), and Random Antenna Array (RAA) to address this problem in terms of typical antenna models. Antenna gain can be rewritten again in terms of array factor of *N* sensors as
(5)$$G\left(\theta ,\phi \right)=\frac{AF^{\kappa }{\left(\phi /r,\theta \right)}^{2}}{\frac{1}{4\Pi }\int_{0}^{2\Pi }\int_{0}^{\Pi }AF^{\kappa }{\left(\phi /r,\theta \right)}^{2}\sin\left(\theta \right)d\theta d\phi }$$

### 3.1. Uniform Linear and Mesh Arrays

Beamforming is achieved by applying phase shift to each element in the array such that its mainbeam points towards the desired direction. Assume far-field conditions, where the information regarding the locations of collaborative sensors are shared, then the resultant array factor when the beams from collaborative sensors are directed towards a base station with azimuthal angle ϕ∈[−π,π] away from the cluster head is given as follows:(6)$$AF^{\kappa }\left(\phi /r,\omega \right)≜\sum_{\kappa =1}^{N}\omega_{\kappa }\exp^{-ȷ\frac{2\pi }{\lambda }r_{\kappa }\left[\cos\left(\theta -\psi_{\kappa }\right)\right]}$$
where $\omega =[\omega_{1},\omega_{2},\ldots ,\omega_{N}]$, and $\omega_{\kappa }$ is κth sensor’s transmission weight, defined as
(7)$$\omega_{\kappa }=\xi_{\kappa }\exp^{ȷ\alpha_{\kappa }}$$
where $\omega_{\kappa }$ is the amplitude and $\alpha_{\kappa }$ is the initial phase of a κth collaborating sensor is $\alpha_{\kappa }=\frac{2\pi }{\lambda }r_{\kappa }\left[\cos\left(\phi -\psi_{\kappa }\right)\right]$. For ULA, each element in the array is placed along with distance *d* between adjacent elements. The array factor can be expressed as:(8)$$AF^{\kappa }\left(\phi /r,\theta \right)≜\frac{\sin(\frac{N\psi (\phi )}{2})}{\sin(\frac{\psi (\phi )}{2})}$$
where ψ(ϕ) is the phase difference between adjacent elements in the direction ϕ, which is related to *d* by $\psi \left(\phi \right)=\frac{2\Pi }{\lambda }\cos\left(\phi \right)+\delta$, where δ is the progressive phase shift of adjacent elements, due to the physical deployment of elements. Therefore, the angle of mainbeam is $\delta =-\frac{2\Pi d}{\lambda }\cos\left(\phi_{o}\right)$.

Substituting Equation (Equation 8) into Equation (Equation 5), we can calculate the antenna gain for any azimuthal angle for ULA. Meanwhile, for UMA the antenna gain can be calculated for any azimuthal angle ϕ, by substituting Equation (Equation 6) into Equation (Equation 5).

### 3.2. Random Antenna Array

To steer a beam in the desired direction, each sensor selects a boresight direction θ from a uniform random distribution on [0,2Π), completely independent of other sensors. We are interested in improving the overall connectivity to enable sensors to transmit to a larger distance. However, this increase in transmission range is bounded to a certain direction. Hence, the sensors disconnect the connections to other sensors located nearby and might end up being isolated if the mainbeam is too narrow. To overcome this problem, a CB that uses random beamforming by defining the probability of two randomly selected sensors in random deployment are connected via a multihop path is adopted in this paper. Since the deployment of sensors depends on a random distribution, the direction of any other sensor from a chosen sensor has a uniform distribution as well. Therefore, the antenna gain for any azimuthal angle ϕ being uniformly distributed over [0,2Π) is given as [18]
(9)$$AF^{\kappa }\left(\phi_{T},\phi_{R}/r,\theta \right)≜\frac{1}{{\left(2\Pi \right)}^{3}}\int_{0}^{2\Pi }\int_{0}^{2\Pi }\int_{0}^{2\Pi }{\left(G\left(\phi ,\phi_{T},k\right)G\left(\Pi +\phi ,\phi_{R},k\right)\right)}^{\frac{2}{\alpha }}d\phi_{R}d\phi_{T}d\phi$$
where $G(\phi ,\phi_{T},k)$ and $G(\phi ,\phi_{R},k)$ are the transmit and receive antenna gains, respectively.

## 4. Channel Model

We consider a radio channel that is affected by path-loss and shadowing effects. We intend to calculate the received power $P_{r}$ over the distance $d_{\alpha }$ between the source–destination pair for a given transmitted power, $P_{t}$ using [6]
(10)$$P_{r}=\frac{P_{t}G_{r}G_{t}}{10^{\frac{\omega }{10}}d^{\alpha }}$$

We denote the gains of transmitter and receiver as $G_{t}$, and $G_{r}$, respectively. The signal is attenuated by the path-loss as $d^{-\alpha }$ which is usually bounded as 2≤α≤6 multiplied by the shadowing factor ω in dB, which is a Gaussian random variable with zero mean and standard deviation σ(dB).

To quantify these variations, we calculated the attenuation among collaborating and connectable sensors denoted by β. The expected value of $P_{r}$ increases as ε×N, where ε is defined as a function of the phase distribution 0≤ε≤1 [41]. When there is no phase error meaning that beamforming with *ℵ* sensors gives a power gain over transmission as with single element as $E[P_{r}=ℵ]$ [42]. Thus, any improvement or degradation is caused by the distribution of ε. As long as keeping the distribution of connectivity of collaborative sensors is contained in such a way as to keep ε≈1, large gains can be realized using distributed beamforming. The power attenuation between two collaborating sensors denoted by β is calculated as
(11)$$\beta =\frac{P_{t}}{P_{r}}=\frac{10^{\frac{\omega }{10}}d^{\alpha }}{G_{r}G_{t}}$$

This means that two collaborating sensors are successfully connected if the power attenuation is not greater than a threshold $\beta \left(d_{\max}\right)<\beta_{0}$. According to [6], if we replace β in Equation (Equation 10) with $\beta_{0}$, the maximum transmission range $d_{max}$ is given as $d_{max}=\sqrt[{\alpha }]{\frac{G_{r}G_{t}\beta_{0}}{10^{\frac{\omega }{10}}}}$. From Equation (Equation 10), the probability of having no direct connection between two sensors separated by $d_{\max}$ is given by
(12a)$$P(\beta \geq \beta_{th})=P(\frac{d^{\alpha }}{G_{r}G_{t}}\geq \beta_{0})$$
(12b)$$=P({\left(\beta_{0}G_{t}G_{r}\right)}^{\frac{1}{\alpha }}\leq d)$$

The sensor can communication with all its randomly distributed neighbors within a radius *R* given by
(13)$$R={\left(\beta_{0}G_{t}G_{r}\right)}^{\frac{1}{\alpha }}$$

Substituting Equation (Equation 13) into Equation (12), we get $P\left(\beta \geq \beta_{0}\right)=P\left(R\leq d_{\max}\right)$. From this analysis, we can derive the first constraint in Equation (Equation 14) of the optimization problem which depends on the antenna gains of each connected source–destination pair. The constraint can vary with the number of connectable and collaborative sensors with different directions since $G_{r}$ and $G_{t}$ are also affected and vary in different directions [40].
(14)$$0≨d_{max}≨d_{\kappa }$$

## 5. Connectivity Model

Let the collaborative sensors be considered to be input variables $\chi =[x_{1},x_{1},x_{1},\ldots ,x_{\kappa }]$ assumed to be Gaussian distributed with density ρ over a given area in a 2D plane. Each sensor first calculates the approximate number neighbors after the deployment by exchanging the information among the sensors in the network. The expected number of sensors in an area I can be calculated as $S=\rho \pi d_{max}$. However, the output function of the area as $f_{N}\left(ℵ\right)=\frac{\rho }{\kappa }$, in the range κ∈0≤ℵ≤N of each of the *ℵ* of *N* grouped by clusters of collaborative sensors of probability of size $\frac{1}{ℵ}$. Therefore, the expected number of collaborative sensors in each cluster is given by $ℵ=\frac{S}{N}$.

Linear beamforming technique implies that every sensor directs its mainbeam towards the geometric center of the network, where the connection can be easily established if any two sensors are aligned within each other’s mainbeam. Hence, we can say that every two sensors are facing each other if their main beams are pointing towards each other or, more precisely, if they are deployed within the width of each other’s main beam.

We expect that the sensors near the base station have high probability of being connected together via their mainbeams. The size of the connectivity area is proportional to the gains of the mainbeam. Thus, the radiated power is fixed at each sensor, where is an ideal distributed beamforming with fixed number of collaborative sensors results in maximum gain which is approximately equal to $N^{2}$ fold whereas the expected value of $P_{r}$ at the base station increases as *N* increases [43].

On the other hand, the probability of two sensors facing each other in the random deployment is independent of the distance from the base station. Therefore, the connectivity performance of random beamforming is almost the same throughout the network. The gain of mainbeam is independent of the boresight direction; it is always equal to the number of sensors *N*. As *N* increases, the mainbeam of each sensor can reach a greater distance from the neighbor, while its width stays the same. Collaborative beamforming concentrates the radiated power in a certain direction toward the base station. Moreover, CB distributes power consumption overall sensors when the collaboration involves ${ℵ}^{\kappa }$ sensors up to an $N^{\kappa }$ [41]. The gain $G_{r}$ is defined as the maximum in mainlobe depending on the network topology as well as the minimum value of an $\frac{1}{N^{\kappa }}$ fold with the reduction in the received power everywhere else [41]. An example of IoT application is monitoring rural areas by deploying sensors. The locations are considered random due to wind, releasing mechanism, speed, and height of the local environment. The array factor is written in terms of the weighted sum of the individual elements as follows
(15)$$AF^{\kappa }\left(\phi /r,\phi \right)≜\int_{\kappa =0}^{ℵ-1}\omega_{\kappa }\left(\vec{\chi }\right)\exp^{ȷ\varphi_{\kappa }}\exp^{-ȷ\frac{2\pi }{\lambda }d{\left(\vec{\chi }\right)}_{\kappa }\left(\phi \right)}$$

This describes the spatial response of *ℵ* element array factor. The positions are denoted as $\vec{\chi }=\left\{x_{\kappa },y_{\kappa }\right\}$, κ ranges from 0 to ℵ−1, and each element has a weight $\omega_{\kappa }\left(\vec{\chi }\right)$ defined as
(16)$$\omega =\int_{\kappa =0}^{ℵ-1}\omega_{\kappa }\left(\vec{x}\right)\exp^{-ȷ\alpha_{\kappa }\left(\vec{x}\right)}$$

The randomness of positions and the number of sensors cause variations in the respective beampatterns, whereas both the sidelobes’ positions and magnitude are affected by the network topology. To ensure the connectivity among these collaborative sensors, the Cartesian product for each cluster grouped as ${ℵ}^{\kappa }$ of probability size ${ℵ}^{-\kappa }$ is used. Each cluster of connected collaborative sensors can be labeled by a set of *ℵ* coordinates ${\vec{x}}_{ı}=\left\{\left(x_{ı1},y_{ı1}\right),\left(x_{ı2},y_{ı2}\right),\ldots ,\left(x_{ı\kappa },y_{ı\kappa }\right)\right\}$, whereas $\left(x_{ıȷ},y_{ıȷ}\right)$ defines the components of one cluster denoted as ${\vec{x}}_{ı}$ of collaborative connected sensors. The Latin hypercube sample that is denoted as $h(\vec{x})$ is used to obtain a random selection of *ℵ* of clusters ${\vec{x}}_{1},{\vec{x}}_{2},\ldots ,{\vec{x}}_{ℵ}$ conditioned by the probability of sensor connection for each component of a cluster. The set ${\left\{{\vec{x}}_{ıȷ}\right\}}_{ı=1}^{ℵ}$ is defined as a permutation of the integers 1,…,N. Moreover, the density function of *N* given as [44]
(17)$$N∋\left(x_{ıȷ},y_{ıȷ}\right)=\left\{\begin{array}{cc} 1, & \mathrm{if}\;{ℵ}^{\kappa }f\left(\vec{x}\right)⟷\left(x_{ıȷ},y_{ıȷ}\right)∋{\vec{x}}_{ℵ}\\ 0, & \mathrm{otherwise} \end{array}\right.$$

The distribution of connected sensors is seen to be the measurement of the local network connectivity (i.e., the connectivity of collaborative sensors).

**Definition** **1.**
*The probability of connectivity P(conn) is the probability that for each node in a network, there is connection to any other node.*


The probability of a sensor connects with any of its neighbors is an important metric to evaluate the local network connectivity. Moreover, the distribution of connected sensors $P_{\vec{x}}\left(conn\right)\leq d_{max}$ follows Gaussian process is given as
(18a)$$P_{\vec{x}}\left(conn\right)\leq d_{max}=\sum_{\forall {\left\{{\vec{x}}_{ıȷ}\right\}}_{ı=1}^{ℵ}}P_{\vec{x}}\left(conn|N∋{\left\{{\vec{x}}_{ıȷ}\right\}}_{ı=1}^{ℵ}\right)P\left(N∋{\left\{{\vec{x}}_{ıȷ}\right\}}_{ı=1}^{ℵ}\right)$$
(18b)$$=\sum \int_{\left\{{\vec{x}}_{ıȷ}|h\left(\vec{x}\right)\leq \exp^{-E[℘]}\right\}}f\left(\vec{x}\right)d\vec{x}\left(\frac{1}{{ℵ}^{\kappa }}\right)$$
(18c)$$=\int_{h\left(\vec{x}\right)\leq \exp^{-E[℘]}}f\left(\vec{x}\right)d\left(\vec{x}\right)$$

Where *℘* denotes the node degree of connectivity, which is defined as the number of sensors such that any given sensor can connect with directly, and E[·] denotes the expectation operator. Thus, the average sensor degree can be expressed as $E\left[℘\right]=\rho \pi \left[d_{max}\right]$. To present the variance of the Latin hypercube sampling, it is usually not possible to obtain closed-form expressions for Equation (Equation 17), an approximation is thus required. We establish this by means of Latin hypercube sampling in the coverage area where the collaborative sensors are deployed. We use an indictor denoted as ε with
(19)$$\varepsilon_{{\vec{x}}_{ıȷ}}=\left\{\begin{array}{cc} 1, & \mathrm{if}\;\left\{\left(x_{ı1},y_{ı1}\right),\left(x_{ı2},y_{ı2}\right),\ldots ,\left(x_{ı\kappa },y_{ı\kappa }\right)\right\}\leq d_{max}\\ 0, & \mathrm{otherwise} \end{array}\right.$$

It is shown that the probability of connectivity is influenced by shadow fading and antenna gain, which depends on the position and distance among the selected sensors for each cluster with boundary defined as $I=[\vec{x}-f_{N},\vec{x}+f_{N}]$, where I depends on the point $\vec{x}$ of interest and limits $f_{N}$, which defines the boundaries of the coverage area. Therefore, we considered the randomly directed antenna for the cluster of sensors selection scheme. Each sensor can randomly select its main beamforming direction and then progressively change its direction based on robustness of particle swarm optimization by acquisition of the pre-knowledge of locations of all its neighbors. In other words, starting with initial random weights, the selection of sensors can refine the weights based on sharing the information among them. Figure 4 shows the relative positions of collaborative sensors, where *d*, and $\phi_{0}$, $\theta_{0}$ denote the distance and angle between two connected sensors, respectively. All sensors with angles are Gaussian distributed in (0,2π].

## 6. Sidelobe Reduction via Optimizing the Sensor Selection Algorithm

The consideration of taking random samples of sensor locations provides additional degree of freedom for controlling the sidelobes. To achieve the desired mainlobe accompanied by reduction of the sidelobes, it is required to select a subset of connected and collaborative sensors from candidate sensors within the same coverage area of each κth source–destination pair.

This sampling is assumed to be as Latin distributed random variable which represents the whole coverage area considering the fluctuations/shadowing effects in the channel. It depends on the distance among the selected collaborative sensors, whereas the sensors are close to each other, while the base station is located far from these selected collaborative sensors. Moreover, the network can be viewed as homogeneous and the power attenuation due to different paths are unequal. Therefore, we can either adjust the gains of the receivers or the power/the number of the collaborative sensors participating in CB by adjusting the weights.

### 6.1. The Sensor Selection Algorithm

Node selection procedure works on the basis that the positions of the sensors forming an antenna array are selected such that their mutual phase offset due to their locations creates a coherently combined signal at the cluster head. Consequently, the node selection algorithm involves carefully selecting the best position for each collaborative sensor and then positioning according to the logical connectivity such that the beampattern is optimal. Assume ${ℵ}^{\kappa }$ be a set of collaborative sensors to be selected from $N^{\kappa }$, i.e., ${ℵ}^{\kappa }\subset N^{\kappa }$, to form beampattern towards the base station. The path connectivity assigns a set of collaborative sensors to each source–destination pair that forms a specific topology. Each cluster consists of many sensors, thus the mainlobe of the beampattern is unstable for different number of ${ℵ}^{\kappa }$ of different subsets $N^{\kappa }$. This occurs as long as the coverage area does dynamically change according to IoT environment as well as the requirements of the applications in that environment. It is necessary to take into consideration these two conditions to maximize mainlobe towards the intended base station and minimize the sidelobes.

We have developed an algorithm for sensor selection based on robust optimization scheme satisfying certain objective functions. First, to find an optimal solution for uncertain objective function which guarantees that the sidelobe levels at the direction of intended base station are below a certain prescribed value. Second, to find an optimal solution that is stable under fluctuations of the channel environments.

However, it seems a challenge to find a robust optimal solution for the objective function $f\left(\vec{x}\right)\to \max\left(mainlobe\right)/\min\left(sidelobe\right)$ on a continuous optimization problem with uncertainty fluctuations in the collaborative sensors. This is because of the difficulty of realizing the gain achieved from the fluctuations in input variables to get an optimal solution. Hence, to assess the robustness of a solution for the objective function, there are common measures to calculate the expected fitness function taking into account the input variable fluctuations and their probability of deployment via a probability distribution function $pdf(\vec{x})$ over the whole input variable space ${\left[{\vec{x}}_{ı}=\left\{\left(x_{ı1},y_{ı1}\right),\left(x_{ı2},y_{ı2}\right),\ldots ,\left(x_{ı\kappa },y_{ı\kappa }\right)\right\}\right]}^{ℵ\subset N}$, where ℵ⊂N is the problem’s dimension.

As mentioned in the previous section, the spacing among the collaborative sensors affects the beampattern shape of the antenna array. Thus, selecting random sensors positions i.e., uncertain sensor positions may lead to positions errors. However, the minimum SLL of an antenna array with fixed and selected sensors positions is higher than that of the antenna array with randomly selected sensors if the number of antenna array is the same, since the distance among the randomly selected sensors has maximum Euclidean distance corresponding to a certain hop-distance which is considered to be a practical metric for modeling spatially random sensor network. Specifically, the connectivity in terms of estimating the area of interest is defined by maximum distance that can be covered in multihop paths. Furthermore, the maximum Euclidean distance is directly related to the estimated hop-distance which is equal to the least number of hops overall multihop paths between any two locations. Therefore, CPSO is used for the optimization of sensor selection whereas the topology of random selection becomes regular topology such as linear, mesh, tree, or ring and the SLL deteriorates further. The objective function is given as
(20)$$Z=\int_{\kappa =1}^{h\left(\vec{x}\right)\leq \exp^{-E[℘]}}f\left(\vec{x}\right)pdf\left(\varepsilon_{{\vec{x}}_{ıȷ}}\right)d\left(\varepsilon_{{\vec{x}}_{ıȷ}}\right)$$

The calculation of the objective function takes into consideration the number of distributed collaborative sensors sampled and the possibility of connectivity via a probability distribution function of deployment over the coverage area ${\left[{\vec{x}}_{ı}=\left\{\left(x_{ı1},y_{ı1}\right),\left(x_{ı2},y_{ı2}\right),\ldots ,\left(x_{ı\kappa },y_{ı\kappa }\right)\right\}\right]}^{ℵ\subset N}$, where ℵ⊂N is the problem’s dimension. The required number and topology of distributed sensors *N* must first be decided to construct the antenna elements *ℵ* in the network.

Consider *N* as the total number of sensors, the number of collaborative sensors to be selected is ℵ⊂N, where ℵ≤N, and the number of distributed collaborative sensors sample of one active cluster to be tested in each iteration be κ≤ℵ. The number of distributed collaborative sensors samples for achieving an optimal solution is an important factor, since each selected sensor must share the transmission range among others within the active cluster. This implies that the characteristic parameters of a CB of the selected sensors must maintain their stable values as the number of samples increases. Moreover, while testing one group or a group of sensors, we need to check if beamforming of the corresponding distributed collaborative sensors sample increases the mainlobe and decreases sidelobes in the intended and unintended direction(s), respectively. Thus, the minimum number of these samples can be determined when the mainlobe and sidelobes reach their stable values. To avoid occurrence of grating lobes, the interspace among sensors should satisfy $r_{o}\leq \frac{\lambda }{2}\leq \rho \pi d_{\max}$ [20]. The selection process can be summarized as follows:Step 1Each sensor starts to explore the neighbors in all directions using randomly directed antenna scheme. Assuming that sensors in the cluster know their distance relative to the cluster head by directly communicating with the base station, then the weights of the signal transmitted and received by the cluster head can be computed.Step 2Suppose that the cluster head and its neighbors are arranged in a specific network topology such as linear, mesh, or random. Then, the cluster head is at approximately equal distances from each other in linear and mesh topology, meanwhile it is not equal in random topology. Hence, assume that $\vec{x}\left(r_{ı,\phi_{ı}}\right)$ acts as the reference of the active cluster to select the collaborative sensors to perform the CB. We need to check the closest selected collaborative sensors ${\vec{x}}_{C}\left(r_{ı,\phi_{ı}}\right)$ within transmission range of cluster head as
(21)$$d_{{\vec{x}}_{CB}}\left(r_{ı,\phi_{ı}}\right)=\min\left\|d_{\vec{x}}\left(r_{ı,\phi_{ı}}\right)-d_{{\vec{x}}_{C}}\left(r_{ı,\phi_{ı}}\right)\right\|$$Step 3Sensors are selected to form an antenna array that adjusts its own radiation pattern to radiate the beampattern in certain direction and reject the signals from other directions. This is controlled by the weighted signal of the collaborative sensors which are computed and the signal is transmitted to the cluster head based on robust particle swarm as discussed in Section 6.2 to satisfy the objective function defined in Equation (Equation 20). The memetic objective function shows that these weights could be adapted to change based on the signal environment in terms of fluctuations/shadowing in the channel. Therefore, the optimal radiation pattern can be obtained through maximizing the mainlobe and minimizing the sidelobes by perturbing the positions of selected collaborative sensors in the antenna array.

### 6.2. The Robust Canonical Particle Swarm

PSO is an iterative, population-based intelligent computational algorithm which searches for optimal solution for non-linear continuous problems. The performance of PSO depends on the power of the solutions’ population that endorse it to find several optimal solutions, i.e., particles over multiple generations. Therefore, one improvement would be to generate randomizing potential particles and continue using the dynamic distances among the particles to dictate communications with selected particles. However, such as technique requires intensive computations to determine these distances, and therefore, PSO is not viable for larger dimensions for the following reasons.
PSO can struggle as it can be difficult for the combination of the number of particles, i.e., swarm to enlarge the search space size to converge on an optimal solution, often getting stuck in local optimal.The particles are connected as a topology which enables several communication paths among its members and the way the swarm is searching the landscape. Since the neighborhood topology changes the pattern of the swarm, convergence and diversity differ from one topology to another. In ring topology known as local PSO $p_{best}$ the information is slowly distributed among the particles. Therefore, using the ring topology slows the convergence rate because the optimal solution must propagate through a few neighborhoods before affecting all particles in the swarm. Meanwhile, mesh topology uses a kind of fully connected topology that is known as global PSO $g_{best}$. The communication among the particles is expedition, and swarm quickly moves towards the best solution. Because of the cost of neglecting part of the search space, the population may fail to explore outside of local regions causing the swarm to be trapped in local optimal solution.PSO suffers from the dual problems of outdated memory due to the environment dynamism and diversity loss, due to convergence.PSO does not always work perfect and may need tuning of its behavioral control parameters such as weight inertia and constriction factor.

The following points clarify the differences between PSO and CPSO:Particles in CPSO possess memory of the best location visited in $p_{best}$ and its fitness value.Every particle shares information with every other particle in the swarm and there is a single $g_{best}$ attractor representing the best location of the entire swarm.CPSO provide more control to parameters of the swarm behavior by mathematical operators such as ⊗, ⊕, and ⊙ for increasing and decreasing the inertial weight and velocity clamping.

The PSO algorithm starts as follows: an initial swarm of particles as one-dimensional vector $\vec{x}=\left[x_{1},x_{2},x_{3},\ldots ,x_{n}\right]$ are generated at random. These particles communicate and move in the research space to allocate optimal solution and reach at optimal area. The position of particles is changed by adding a velocity to the current position: $\vec{x}\left(t+1\right)=\vec{x}\left(t\right)+\vec{\upsilon }\left(t+1\right)$. In comparing PSO with CPSO, the proposed optimization algorithm exists as a swarm of particles, i.e., sensors and each particle *ı* resides at position $\vec{x_{ı}}$. These particles move with a certain velocity $\vec{\upsilon_{ı}}$ over the search space. Each position is associated with a fitness value given by the objective function during several iterations of evaluations ι. For *N*-dimensional problem, the position and velocity can be specified by ℵ×N matrices as follows
(22)$$\vec{x}={\left(\begin{array}{cccc} x_{11} & x_{12} & \ldots & x_{1|N|} \end{array}\right)}^{x_{\left|ℵ||N\right|}}$$
(23)$$\vec{\upsilon }={\left(\begin{array}{cccc} \upsilon_{11} & \upsilon_{12} & \ldots & \upsilon_{1|N|} \end{array}\right)}^{\upsilon_{\left|ℵ||ℵ\right|}}$$

As mentioned, *ℵ* defines the number of particles/sensors in the swarm. Each row of the position matrix represents a possible solution to the optimization problem after evaluating the objective function $f(\vec{x})$ multiplied by connectivity indicator factor $\varepsilon_{\vec{x}}$ defined as follow
(24)$$\varepsilon_{\vec{x}}={\left(\begin{array}{cccc} \varepsilon_{\vec{x_{11}}} & \varepsilon_{\vec{x_{12}}} & \ldots & \varepsilon_{{\vec{x}}_{1|N|}} \end{array}\right)}^{{\vec{x}}_{\left|ℵ|ℵ\right|}}$$

Hence, the best position is defined as the local or personal-best ${\vec{x}}_{p_{best}}^{\iota }$ achieved by the evaluation of the objective function for collaborative sensors during the first iteration of selection as follows.
(25)$$p={\left(\begin{array}{cccc} p_{11} & p_{12} & \ldots & p_{1|N|} \end{array}\right)}^{p_{\left|ℵ|ℵ\right|}}$$

Moreover, each particle knows its ${\vec{x}}_{p_{best}}^{\iota }$ best value so far, which corresponds to experience of each particle. Among all iteration of all collaborative clusters, a new position is considered to be a global-best position ${\vec{x}}_{g_{best}}^{\iota }$ and is defined by
(26)$$G=[g_{1},g_{1},\ldots ,g_{\left|ℵ\right|}]$$

Moreover, each particle may either achieves a better position or does not by adjusting its velocity based on the evaluation of the objective function defined in Equation (Equation 20) in the previous position as well as the best previous positions of all particles. Specifically, each particle tries to adjust the velocity based on the following information to modify its position and refine its own weight:The distance between the current position and the ${\vec{x}}_{p_{best}}^{\iota }$The distance between the current position and the ${\vec{x}}_{g_{best}}^{\iota }$

After a certain number of iterations, ϵ all particles of the swarm converge towards positions that are optimal either locally or globally. The degree of influence among the personal and global-best solutions should be defined by coefficients. For instance, the personal position is defined by coefficient $\phi_{1}$, meanwhile the influence of the best global or neighborhood exchange solution is defined by a coefficient $\phi_{2}$. Accordingly, updating the velocity and position of an κth particle at ϵth iteration is defined mathematically as
(27a)$${\vec{\upsilon }}_{ı}^{\iota +1}:=[\omega *\upsilon_{\kappa }^{\iota }]$$
(27b)$$+[\phi_{1}*rand\left(\right)⊗\left({\vec{x}}_{p_{best}}^{\iota }-{\vec{x}}_{\kappa }^{\iota }\right)]$$
(27c)$$+[\phi_{2}*rand\left(\right)⊗\left({\vec{x}}_{g_{best}}^{\iota }-{\vec{x}}_{\kappa }^{\iota }\right)]$$

The position of the κth particle/sensor for the next iteration is updated such that ${\vec{x}}_{\kappa }^{\iota +1}={\vec{x}}_{\kappa }^{\iota }+\upsilon_{\kappa }^{\iota +1}$, where $\phi_{1}$ is the cognitive parameter, $\phi_{2}$ is the social parameter, ω is defined as the inertia weight index that tries to explore a new cluster, and rand() is a random number within [0,1]. The ratio of $\phi_{1}:\phi_{2}$ determines the importance of ${\vec{x}}_{p_{best}}^{\iota }$ and to ${\vec{x}}_{g_{best}}^{\iota }$. There are many mays to implement the constriction coefficient ξ. One of the simplest methods is shown in Equation (27), whereas the first term of expresses the previous velocity of the information vector (in other word the personal-best vector before starting to change particles/sensors information with their neighbors). Meanwhile, the second and third terms are used to change or update the velocity of the information vector from personal-best vector to global-best. Without the second and third terms, the vector will keep on moving in the same direction until it hits the boundary. The inertia weight ω, regulates the influence of the previous velocity $\upsilon^{\iota }$ on the new velocity $\upsilon^{\iota +1}$, and it should not be confused with beamforming coefficient weights, whereas the inertia weight is a constriction coefficient, which helps to balance global exploration and local exploitation, and thus helps in preventing velocity explosions. It is defined as follows [41]:(28)$$\omega =\frac{2}{\phi +\sqrt{\phi^{2}-4\phi }},\;\mathrm{with}\;\phi =\phi_{1}+\phi_{2}>4$$

The Clerc’s constriction method is used [45], where ϕ is commonly set to 4.1 which helps to control the convergence of particles, whereas $\phi_{1}=\phi_{2}=2.05$, and ω is set as a constant value. Therefore, the constricted particles converge without using any velocities values at all, i.e., removal of velocity clamping facilitates larger exploration abilities of the swarm. This implies a swarm behavior that is eventually limited to a small area of the feasible search space containing the optimal solution. However, Equations (27) and (Equation 28) are formulated such that the CPSO has no problem-specific parameters addressing all efficiency of exhaustive search over all possible solutions. Indeed, a logical connectivity describes how directional beamforming affects a network topology and how the information is shared and transmitted by one sensor to another. These topologies are created by orientating the mainlobe direction of the selected collaborative sensors based on sharing and exchange information of the locations from their neighborhood towards the geometric center of the cluster. Hence, these topologies may have prefect and imperfect information of the position of other sensors. According to Equation (Equation 21), the topology of the selected collaborative sensors as shown in Figure 5 will enable the CPSO to have a higher diversity and then being able to find global optimal solution better than all topologies as shown in Figure 6 which are well suited for finding local optimal solution. After a certain number of iterations ι, a custom topology creates group of collaborative sensors with a maximum mainlobe and minimum sidelobe by maximum of κ out of *ℵ* sensors or particles in total. It should be noted that the purpose of using the present swarm topology is to combine the benefits of all type of topologies in terms of linear, mesh, and tree. We rewrite the objective function as
(29)$$\max Z=\int_{\kappa =1}^{h\left(\vec{x}\right)\leq \exp^{-E[℘]}}f\left(\vec{x}\right)pdf\left(\varepsilon_{{\vec{x}}_{ıȷ}}\right)d\left(\varepsilon_{{\vec{x}}_{ıȷ}}\right)$$
subject to
(30a)$$\begin{array}{c} 0≨d_{max}≨d_{\kappa }, \end{array}$$
(30b)$$\begin{array}{c} \varepsilon_{{\vec{x}}_{ıȷ}}=\left\{\begin{array}{cc} 1, & \mathrm{if}\;\left\{\left(x_{ı1},y_{ı1}\right),\left(x_{ı2},y_{ı2}\right),\ldots ,\left(x_{ı\kappa },y_{ı\kappa }\right)\right\}\leq d_{max}\\ 0, & \mathrm{otherwise} \end{array}\right. \end{array}$$

The objective function in Equation (Equation 29) can be applied to both isotropic and realistic antenna patterns according to the type of IoT applications as well as their requirements. However, there is no closed-form expression of Equation (Equation 29) for realistic antenna model or even for a simplified antenna model. Furthermore, it is not possible to obtain closed-form expression for the optimization problem hence an approximate solution is needed.

We adopt a popular approach for this approximation by means of Latin distribution of collaborative sensors sampled for the fitness value of the objective function in an area of interest [44]. For the approximation of the maximum/minimum values of the objective function, this means that *n* samples must be taken out of an area of interest I and the max/min of the sampled fitness value would be returned as
(31)$$f\left(\vec{x}\right)=\left\{\begin{array}{cc} \; & \max\{f\left(\vec{x}+\varepsilon_{\kappa }\right)|\kappa \in \left\{1,\ldots ,ℵ\right\}\}\\ \; & \min\{f\left(\vec{x}+\varepsilon_{\kappa }\right)|\kappa \in \left\{1,\ldots ,ℵ\right\}\} \end{array}\right.$$

## 7. Performance Evaluation

IoT applications such as in military, industrial, or smart-city are considered to illustrate the benefits of the proposed CPSO algorithm in determining the complex antenna array weights to best meet a specified far-field beampattern requirements and to reveal the advantages of the sensor selection algorithm for optimizing the beampattern. The proposed algorithm is implemented using MATLAB. Throughout the experiments implemented, we consider initializing the position matrix at the beginning of the experiment as linear, mesh or randomly distributed in a plane of area L×L m^2^. For example, L=1200 m is initiated and then we select the position matrix depending on the Sensor Array Analyzer toolbox of MATLAB to construct and analyze common sensor array configurations in terms different types of network topology. On the other hand, the overall network connectivity depends on various factors such as the channel randomness and the antenna model. Higher shadow fading variance leads to higher network connectivity, since it refers to the random variations of the received signal power around a mean value that is noticed when collaborative sensors are located at a given distance from a base station. Therefore, if we fix one of the other factors such as the directional antenna response in the desired direction to the base station, then we are able to achieve higher network connectivity to analyze the performance of CPSO against the conventional beampattern and the convex optimization in three different network topologies.

The proposed CPSO algorithm tries to perturb (i.e., optimize) the complex weight coefficients and position of each sensor in turn. Indeed, we can say that it is possible that sparse antenna array elements with continuously spaced sensors could have a high degree of freedom in increasing and decreasing the mainlobe and sidelobes, respectively. Thus, the mainlobe of beamforming is included in the objective function Equation (Equation 29) with sidelobe constraints defined in Equation (30). The beampattern of 36, 64, 100, and 256 sensors deployed as equally spaced (uniform distribution) or non-uniformly spaced is calculated for each candidate sensor (i.e., solution) at each cost evaluation. The population is considered with the same number of deployed sensors, meanwhile the number of sampling of distributed collaborative sensors is varied depending on the network topology configuration. The values corresponding to the CPSO are initialized by performing parameters sensitivity test to the best values for $\phi_{1}$, $\phi_{2}$, and ω, whereas the $\phi_{1}=\phi_{2}\in \left[0,4\right]$, and ω∈[0,1.5]. The initialized of parameters are iterated until they finally combined for updating the velocity using Equation (27) which is randomly initialized with values [−1,1] and achieved the optimal solution. Table 2 lists the parameters used in the experiments.

### 7.1. Linear Antenna Array

In this scenario the element locations constitute a ULA of sensors. The first experiment illustrates the beamforming pattern of the antenna array for a linear network topology of 36, 64, 100, and 256 sensors where our objective is to maximize the mainlobe and minimize the sidelobes in the range $\left(0^{{}^{\circ}},360^{{}^{\circ}}\right]$, the corresponding directions are for $\phi =90^{{}^{\circ}}$ with prescribed nulls angles at $\left[30^{{}^{\circ}},50^{{}^{\circ}},100^{{}^{\circ}},180^{{}^{\circ}},220^{{}^{\circ}}\right]$. The beamforming patterns obtained from the CPSO algorithm are shown in Figure 7a–c along with conventional beamforming pattern obtained using the sensor array analyzer toolbox of MATLAB, and the convex optimization explained in [21]. Table 3 summarizes the optimal values obtained using CPSO sensors where the performance is somehow superior than those of convex optimization and conventional linear array.

The CPSO forms a network topology with its objective function given in Equation (Equation 29) constrained by Equation (30) evaluated by investigating the relationship between distance (i.e., the transmission range) and the connectivity obtained by sampling the selected collaborative sensors to achieve a beam towards a specific direction.

It is important to note the effectiveness of the random positioning of selected sensors as solutions in CB to refine their signal weights. Moreover, owing to the arbitrary ⊗ operand process in Equation (27), the optimization of beampattern achieves better improvement by adjusting the velocity towards the best solution so far found by the selected sensor samples. Therefore, it can be seen from Figure 7a that the conventional beamforming compared to convex optimization exhibits traditionally shaped pattern of sidelobe and mainlobe without minimizing and maximizing, respectively. Meanwhile, the CPSO offers an improvement of 18 dB in terms of maximum of the mainlobe and 3 dB to 15 dB in terms of sidelobe suppression. Furthermore, it can also be seen that the beampattern from Figure 7a–c that sidelobes are almost the same for the CPSO, except in Figure 7c,d whereas a little improvement achieved by the convex optimization than CPSO. The implementation presented promising results whereas the efficiency is limited by the exhaustive search of linear network topology with all possible connectivities. This limitation is mainly due to the memory of particles, i.e., the $p_{best}$ and $b_{best}$ and changing the inertia is not being able to move the swarm out of the current optimum when $\upsilon_{ı}\approx 0$, $\forall_{ı}=1,\ldots ,n$. Despite the higher sensor density, the linear collaborative sensors selection has a low probability of connectivity to perform CB and achieve lower sidelobe and higher mainlobe at the angle of $90^{{}^{\circ}}$ compared with CPSO. This gives an explanation that there is a visible difference between the directional and non-directional antenna types when using a simple random sampling of antenna element selection in ULA. Furthermore, this behavior can be explained with boarder effects, i.e., when the sensors are located at the boarder of the coverage area in which their mainbeam(s) may be steered outside the area of interest. Therefore, some of these sensors become isolated from the network and contribute in a negative manner to the connectivity. In summary, the beampattern with ULA yields an improvement of the mainlobe and sidelobes as compared to conventional beamforming and convex optimization in terms of connectivity. This improvement is achieved for several sensors sensor deployed as shown in Figure 7a,b compared to Figure 7c,d in which the improvement decreases with increasing connectivity among a large number of sensors.

### 7.2. Mesh Antenna Array

The second experiment involves a UMA of 36, 64, 100, and 256 elements designed for maximum mainlobe toward $\phi =90^{{}^{\circ}}$ as well as the sidelobe suppression in region $\left(0^{{}^{\circ}},360^{{}^{\circ}}\right]$ and prescribed nulls at $\left[30^{{}^{\circ}},50^{{}^{\circ}},100^{{}^{\circ}},180^{{}^{\circ}},220^{{}^{\circ}}\right]$. Figure 8 shows that broad nulls as deep as −38 dB, −60 dB, and −65 dB are achieved. Table 4 summarizes the optimal values obtained using CPSO where the performance is somehow superior than those of convex optimization and conventional linear array.

Results from the CPSO given in Figure 8a–d depict almost similar beamforming patterns compared to the conventional beamforming patterns. Each sample of beamforming in UMA indicates at first a deterministic and then closely approximates the mainlobe of conventional beamforming pattern achieving the objective function better than convex optimization and below the conventional beamforming pattern. However, this implies that beamforming pattern of each sample does not spread away from the average of conventional beamforming pattern. This is because the proposed selection algorithm based on CPSO acquires a new diversity of solution from outside the area of interest but inside its population solutions by sampling the solution from the $\int_{\kappa =1}^{h(\vec{x})}$ instead of generating a new one from individual or several solutions. Furthermore, the updating velocity in Equation (27) is adjusted based on sampling of solution to guide the future search toward the global solution and then to produce improved samples closer to the optimal solution. We have noticed that the differences in the mainlobe and sidelobe performance among the three algorithms are obvious within increasing the collaborative sensors in antenna array.

On the other hand, the gap of the mainlobe as seen in Figure 8a,b becomes smaller in the three algorithms as illustrates in Figure 8c,d. CPSO shows a clear advantage of selecting collaborative sensors in the reduction of the sidelobe below 0dB which is essential in reducing interference and saving the lifetime of each sensor.

From the results, it is noted that the CPSO overcomes the undesired increment of the sidelobe in conventional beamforming pattern and convex optimization. This arises some questions: Is the increased connectivity among the collaborative sensors that is achieved or not due to these sensors have more neighbors (i.e., have more degree of connectivity *℘*)? Is the increased connectivity path probability obtained from the directional antenna at the higher sensor degree connectivity and causes a higher inference and overhearing among the collaborative sensors?

To address these questions, we concluded that with increasing number of sensors, *℘* increases linearly with increasing κ. In other words, when the beamforming pattern is narrow, the probability of connectivity increases and achieves the optimal results. This is because when directional antennas are used, the probability of connections increases for various distances among the collaborative sensors because of optimizing the mainlobe of the antenna. We believe that the connectivity solutions in the realm of IoT aim not only at supporting the needs imposed by the several practical applications (such as logistics, agriculture, and asset management), but also to support network self-management for these applications. Thus, a part of the use of the performance of CB is to improve the connectivity solutions for a certain application depends on how well its features solve the specific needs of the end application. There are several studies that address the comparison of connectivity solutions for IoT applications by covering many possible measurement domains with relevant parameters for topologies in terms of power or signal strength-based, time-based, and space-based [45,46,47,48].

### 7.3. Random Antenna Array

The third experiment investigates RAA configurations of 36, 64, 100, and 256 elements designed for mainlobe towards $\phi =90^{{}^{\circ}}$ as well as the sidelobe suppression in the region $\left(0^{{}^{\circ}},360^{{}^{\circ}}\right]$ and prescribed nulls at $\left[60^{{}^{\circ}},120^{{}^{\circ}},180^{{}^{\circ}},220^{{}^{\circ}}\right]$. Broad nulls as deep as −5 dB, −8 dB,−18 dB, and −22 dB are observed using the CPSO as seen in Figure 9a–d. Table 5 summarizes the optimal values obtained using CPSO where the performance is somehow superior than those obtain form convex optimization and conventional linear array.

This experiment applies to military applications whereas collaborative sensors are distributed randomly without supporting facilities (such as energy sources or wired) to specified position close to enemy territories. Consequently, their positions may be or not close to the area of interest and the transmission requires high directivity at the desired direction. Multiple sensors are deployed form multihop topology which leads to difficulty in regulating these sensors as a specific topology because of the environmental constraints such as the sensors’ position and connectivity with their neighbors. Hence, the RAA population is formulated next along with its corresponding conventional beamforming pattern and compared with convex optimization as shown in Figure 9a–d, respectively. The probability distribution function for the connectivity factor is then formed amid RAA collaborative sensors located with an area of interest. Given $\int_{\kappa =1}^{h(\vec{x})}$ of the collaborative sensors in random arrangement, the properties of RRA are determined to include the expected beampattern, peak and null locations of the mainlobe and sidelobes. As illustrated in Figure 9a, the proposed CPSO algorithm achieves the most significant suppression of the sidelobes prominent in the regions of $55^{{}^{\circ}}$, $152^{{}^{\circ}}$, and $190^{{}^{\circ}}$, whereas the approximate solution decreases the corresponding level to −5.12 dB, −6 dB, and −23 dB compared to conventional RAA. A higher gain is obtained in mainlobe whereas optimizing the maximum mainlobe is distinguished in the region of $30^{{}^{\circ}},50^{{}^{\circ}},100^{{}^{\circ}},180^{{}^{\circ}},220^{{}^{\circ}}$, whereas the approximate solution provides an increase of 18dB compared to conventional RAA. Figure 9b considers 64 sensors in which the conventional RRA exhibits relatively minimum sidelobe as compared to convex optimization. It can be observed from Figure 9b that the lowest sidelobes are −20 dB, and −88 dB at $55^{{}^{\circ}}$, and $220^{{}^{\circ}}$ for convex optimization and conventional RRA, respectively. However, CPSO significantly minimizes the sidelobe to −60 dB, and −63 dB at $30^{{}^{\circ}}$ and $190^{{}^{\circ}}$, respectively. Different effects are revealed by increasing the number of sensors as shown in Figure 9c,d. The radiation pattern of 100 sensors are given for the conventional RRA showing minimum sidelobe of 3 dB compared to sidelobe of convex optimization of 0 dB at $50^{{}^{\circ}}$, and $150^{{}^{\circ}}$. Meanwhile, the radiation pattern of 256 sensors is shown using the conventional beamforming RRA which exhibits the minimum sidelobe at $65^{{}^{\circ}}$, and $150^{{}^{\circ}}$ compared to convex optimization. In contrast with ULA and UMA, the CPSO achieves an improvement in the maximum of the mainlobe and minimum of the sidelobes for large-scale deployment of sensors compared to conventional beamforming and convex optimization. As concluded in the previous Section 7.1 and Section 7.2, the collaborative sensors are formed based on the size of the transmitted data, given the total number of deployed sensors. Figure 7a and Figure 8a indicate how much sampling is required to optimize the beamforming which is affected by minimum $d_{\max}$ satisfying the distance constraint Equation ([Disp-formula FD30a-sensors-20-02048]). It can be concluded that the capability of CPSO in improving the performance of CB through satisfying the connectivity constraints is revealed when designing practical random antenna arrays in various IoT applications.

The PSO [27], and GA are compared against CPSO and convex optimization [21]. Figure 10 presents the objective function performance comparisons for CPSO, convex optimization, PSO, and GA. The PSO algorithm performs better than GA as shown in Figure 10a. However, in Figure 10b,c GA outperforms the PSO for increased number of sensors and higher iterations. Meanwhile, CPSO proves its superiority in yielding better sidelobe reduction and higher directivity as compared to those obtained by convex optimization, PSO, and GA. This is because the CPSO efficiently selects, samples and computes the κ number of global optimal current excitation weights as well as the number optimal uniform interelement separation for each beampattern of the linear antenna array to arrive at the maximum mainlobe and sidelobe reduction. Table 6 summarizes the optimal values obtained using CPSO for all the antenna arrays considered where the performance is better than those of conventional linear antenna array, convex optimization, PSO, and GA.

## 8. Conclusions

This paper introduces a CPSO optimization algorithm for the synthesis of virtual antenna arrays selected from randomly deployed sensors in the realm of CB for the purpose of optimizing the mainlobe, suppressed sidelobes, and controlled nulls in certain directions. The proposed CPSO algorithm employs a node selection technique to optimize the beampattern of the CB of sensors. Results are compared with corresponding characteristics in the case of uniform and non-uniform sensor distribution geometries taking into consideration the network connectivity. The proposed CPSO can select the active CB sensors and dynamically controlling the beampattern in order to enhance the desired signal while minimizing the sidelobes.

More control of the beampattern is achieved using the proposed CPSO by optimizing not only the positions of sensors deployed, but also the amplitude and phase of excitation applied to each collaborative sensor in the array and exploring other array geometries. The overall conclusion is that the CB using the proposed CPSO provides a better performance compared against conventional beamforming, convex optimization, GA, and particle swarm optimization when the investigation of network connectivity is affected by the number of sensors deployed in the area of interest.

## Figures and Tables

**Figure 1 sensors-20-02048-f001:**
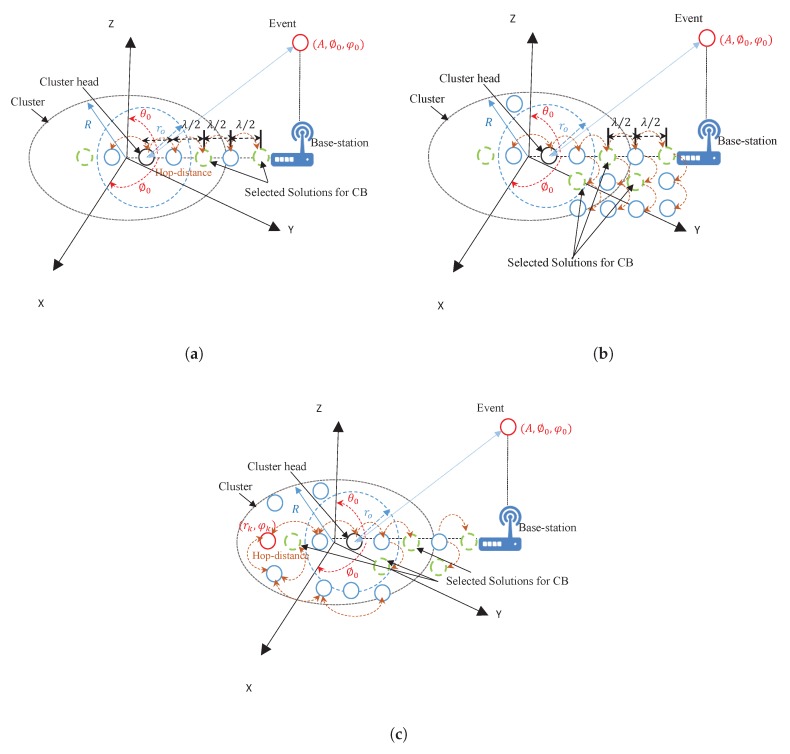
Structure of the system model; (**a**) Uniform Linear Array (ULA), (**b**) Uniform Mesh Array (UMA), (**c**) Random Antenna Array (RAA).

**Figure 2 sensors-20-02048-f002:**
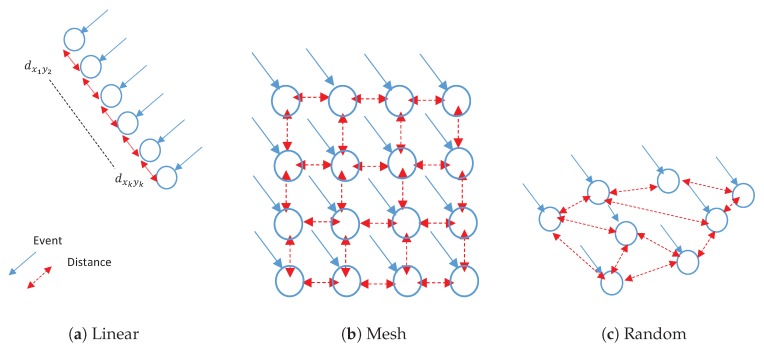
Gain Patterns.

**Figure 3 sensors-20-02048-f003:**
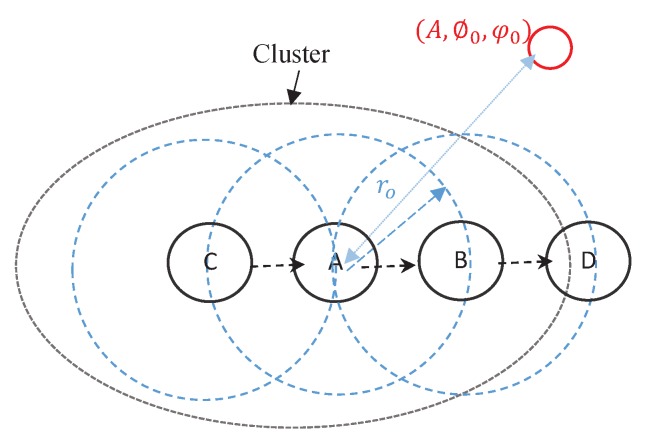
Sensor (A) and sensor (B) can communicate with each other, sensor (C) can communicate with sensor (A). Sensor (B) might connect to sensor (D).

**Figure 4 sensors-20-02048-f004:**
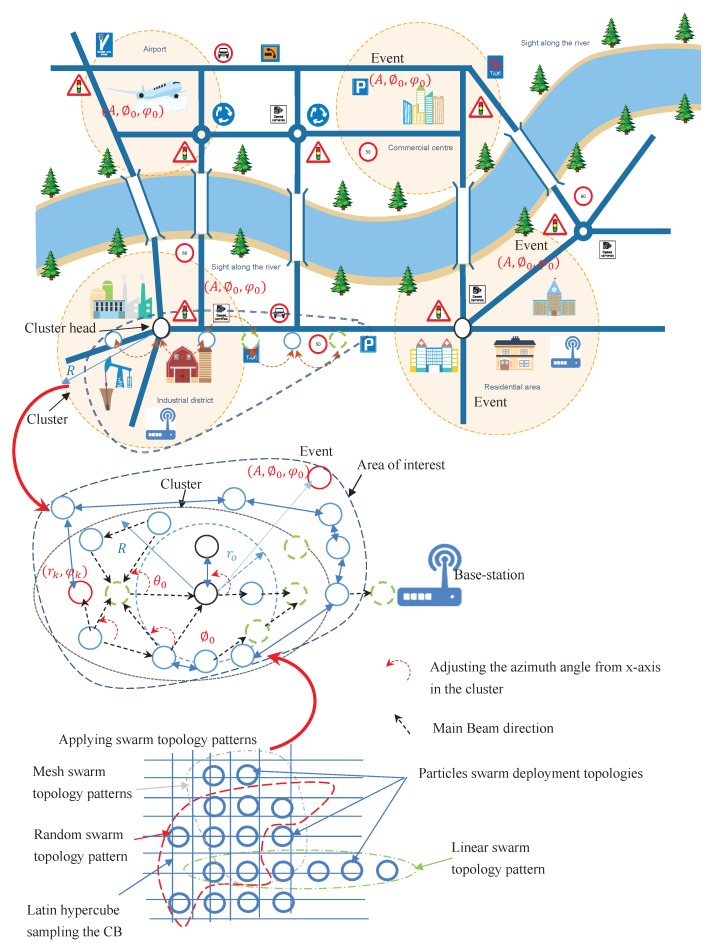
Relative Positions of the Collaborative Beamforming.

**Figure 5 sensors-20-02048-f005:**
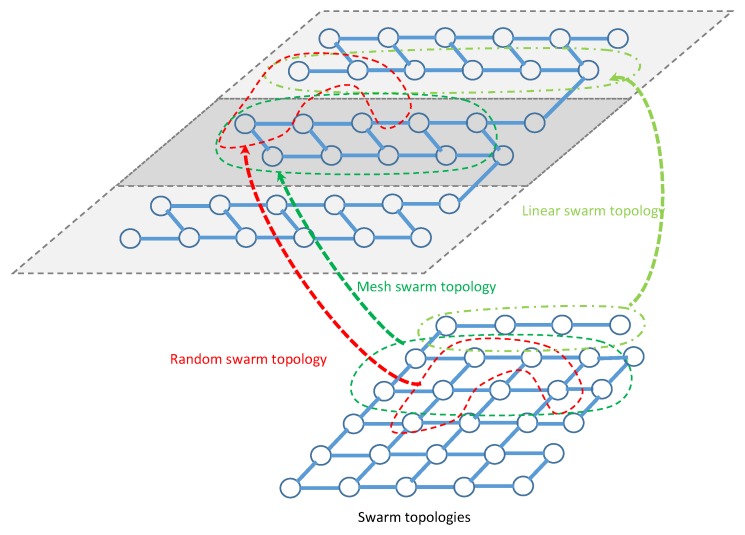
The pattern for sensors deployment with various network topologies.

**Figure 6 sensors-20-02048-f006:**
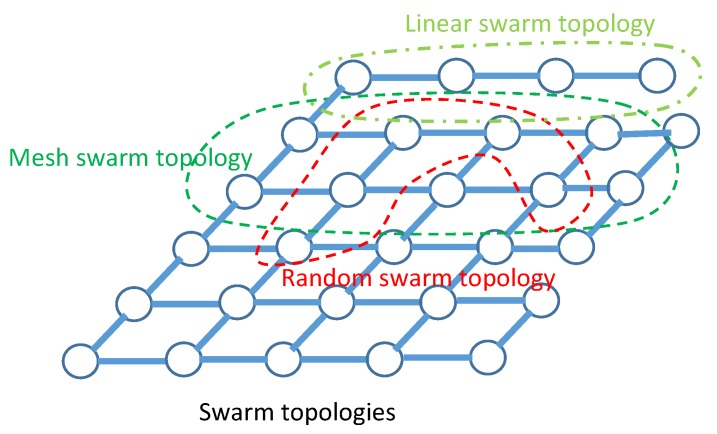
The pattern for swarm deployment with various network topologies.

**Figure 7 sensors-20-02048-f007:**
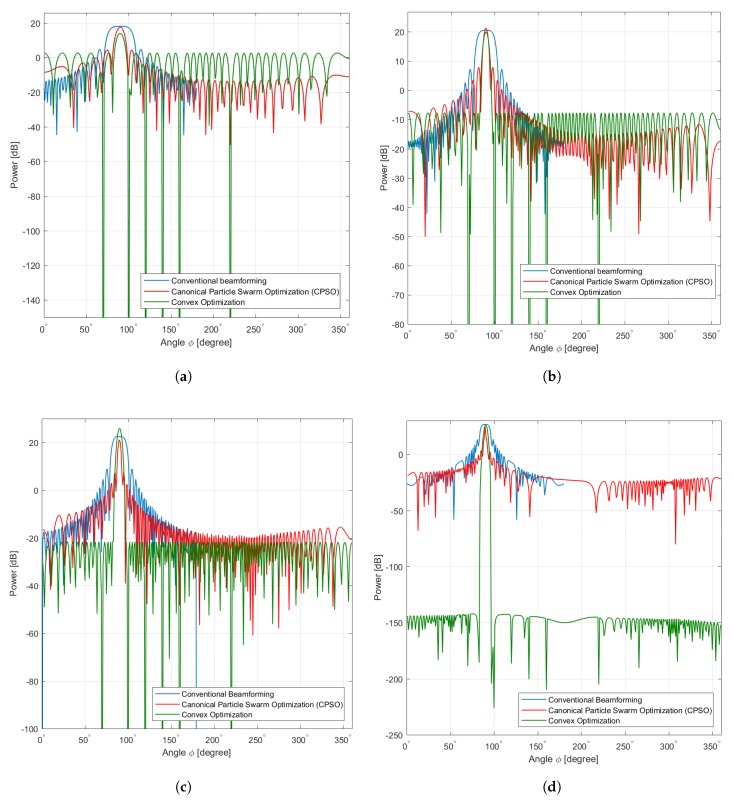
Linear. (**a**) Linear Sensors 36, (**b**) Linear Sensors 64, (**c**) Linear Sensors 100, (**d**) Linear Sensors 256.

**Figure 8 sensors-20-02048-f008:**
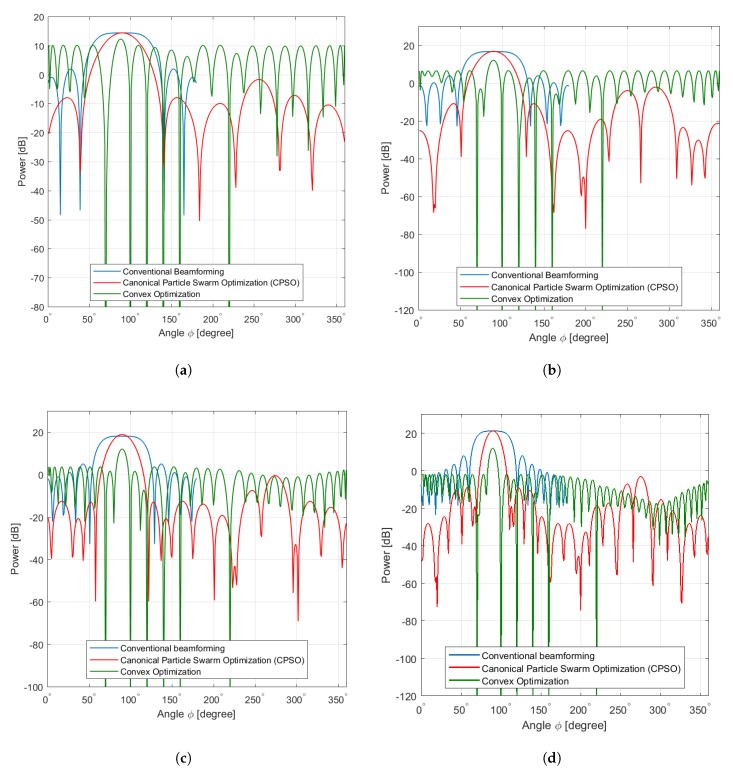
Mesh. (**a**) Mesh Sensors 36, (**b**) Mesh Sensors 64, (**c**) Mesh Sensors 100, (**d**) Mesh Sensors 256.

**Figure 9 sensors-20-02048-f009:**
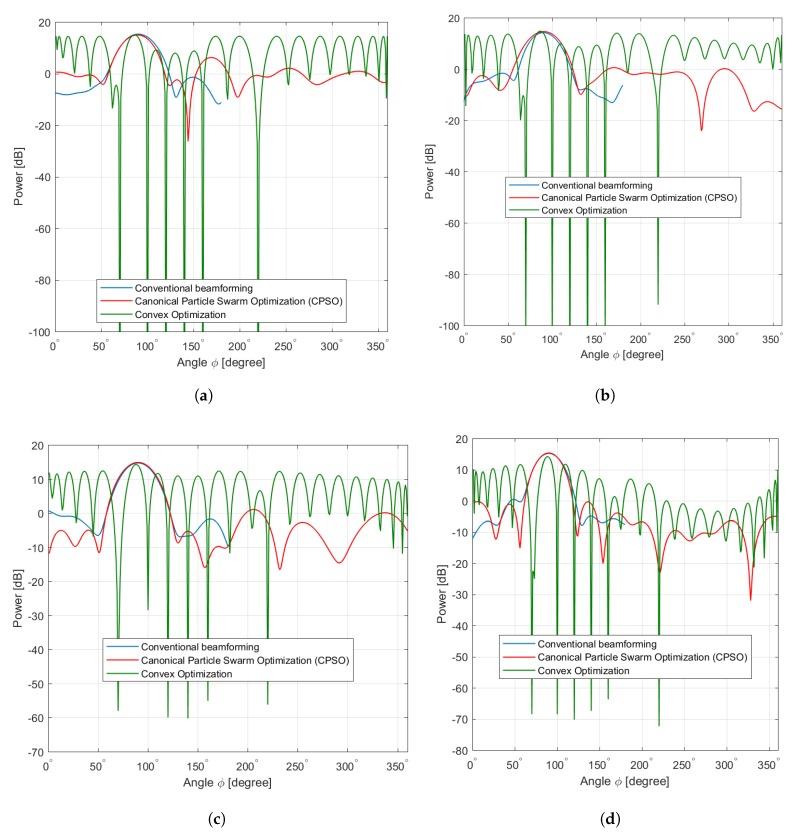
Random. (**a**) Random Sensors 36, (**b**) Random Sensors 64, (**c**) Random Sensors 100, (**d**) Random Sensors 256.

**Figure 10 sensors-20-02048-f010:**
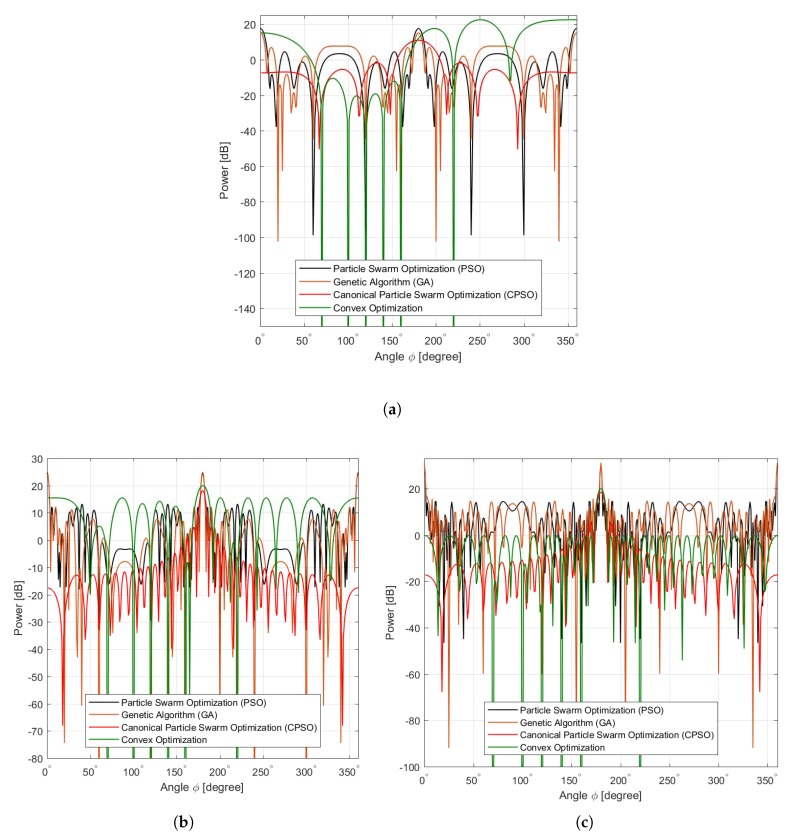
Various Algorithms. (**a**) Mesh Sensors 8, (**b**) Mesh Sensors 16, (**c**) Mesh Sensors 36.

**Table 1 sensors-20-02048-t001:** Notation.

Symbol	Definition
(x,y)	coordinates of deployment sensor over plane
G(V,E)	Graph representation of WSN
(ı,ȷ)	pair of connected sensors
$\left(x_{ı\kappa },y_{ȷ\kappa }\right)$	connected path between any pair of sensors *ı* and *ȷ*
*ℵ*	subset of a group of deployed sensors
κ	number of selected of deployed sensors
*N*	Total number of deployed sensors
*R*	Radius
λ	wavelength
$\varphi_{\kappa }$	initial phase of the κth sensor carrier frequency
$\left(r_{o},\psi_{\kappa }\right),\theta_{o},\phi_{o}$	corresponding coordinate of paired sensors, elevation, and azimuth angle respectively
$d_{\kappa }$	Euclidean distance between κth sensor and any point located in (x,y)
$\left(A,\phi_{o}\right)$	incident coordinate over (x,y) plane
AF	Array Factor of sensors
ω	weight of transmission signal
η	the efficiency factor
ε	define as a function of the phase distribution (the connectivity factor
$\alpha_{\kappa }$	initial phase of κth collaborating sensors
$\zeta_{\kappa }$	complex amplitude of transmission signal
$P_{r}$	receipting power
$P_{t}$	transmit power
$G_{r}$, $G_{t}$	Gain of transmitter and receiver
α	path-loss 2≤α≤6
β	Power attenuation
$d_{\max}$	maximum transmission range between two collaborative sensors
*S*	Density of the deployment sensors
I	the dependent area of interest
*Z*	Objective function
particle	Sensor
$\vec{x}\left(r_{ı,\phi_{ı}}\right)$	Reference of the active cluster
$\vec{x}\left(r_{ı,\phi_{ı}}\right)$	Reference of the active cluster
$\upsilon_{ı}$	Velocity toward selecting optimal solution
$P_{ı}$	Personal-best position for $sensor_{ı}$
$G_{ı}$	Global-best position for $sensor_{ı}$
$p_{best}$	Personal-best
$g_{best}$	Global-best
$\phi_{1}$	Personal-best coefficient
$\phi_{2}$	Neighbor best coefficient
ξ	Constriction coefficient
*n*	The number of samples that must be taken out of area of interest I
$E_{elec}$	The energy dissipation rate to run the radio
$\varepsilon f_{s}$	The one-path model for the transmitter amplifier
$\varepsilon_{mp}$	The multipath model for the transmitter amplifier
ρ	The density of distributed sensors over a given area in a 2D
*℘*	The node degree of connectivity
*ı*	A certain amount of iterations
$h(\vec{x})$	The Latin hypercube sample

**Table 2 sensors-20-02048-t002:** Definition of parameters.

Parameter	Value
The number of sensors	36, 64, 100, 256
Attenuation threshold value $\beta_{0}$	50 dB
Path-loss exponent α	2.5
iterations	200

**Table 3 sensors-20-02048-t003:** Geometry of the linear antenna array consisting of 36, 64, 100, and 256 elements using three different algorithms.

Algorithm	36 Sensors		64 Sensors		100 Sensors		256 Sensors	
	min	max	min	max	min	max	min	max
CPSO	−38 dB	20 dB	−38 dB	20 dB	−20 dB	20 dB	−53 dB	18 dB
Convex	−35 dB	18 dB	−33 dB	19 dB	−39 dB	22 dB	−150 dB	18 dB
Conventional	−20 dB	20 dB	−28 dB	20 dB	−18 dB	21 dB	−22 dB	18 dB

**Table 4 sensors-20-02048-t004:** Geometry of the mesh antenna array consisting of 36, 64, 100, and 256 elements using three different algorithms.

Algorithm	36 Sensors		64 Sensors		100 Sensors		256 Sensors	
	min	max	min	max	min	max	min	max
CPSO	−30 dB	13 dB	−62 dB	18 dB	−55 dB	15 dB	−50 dB	18 dB
Convex	−8 dB	11 dB	−22 dB	15 dB	−22 dB	15 dB	−40 dB	18 dB
Conventional	−47 dB	13 dB	−20 dB	18 dB	−22 dB	18 dB	−15 dB	20 dB

**Table 5 sensors-20-02048-t005:** Geometry of the random antenna array 36, 64, 100, and 256-elements using three different algorithms.

Algorithm	36 Sensors		64 Sensors		100 Sensors		256 Sensors	
	min	max	min	max	min	max	min	max
CPSO	−22 dB	18 dB	−15 dB	18 dB	−11 dB	18 dB	−12 dB	18 dB
Convex	−15 dB	18 dB	−18 dB	18 dB	−25 dB	18 dB	−12 dB	17 dB
Conventional	−7 dB	18 dB	−7 dB	18 dB	−5 dB	18 dB	−8 dB	18 dB

**Table 6 sensors-20-02048-t006:** Geometry of the linear antenna array consisting of 8, 16, and 36-elements using four different algorithms.

Algorithm	8 Sensors		16 Sensors		36 Sensors	
	min	max	min	max	min	max
CPSO	−40 dB	18 dB	−18 dB	18 dB	−38 dB	18 dB
PSO	−38 dB	20 dB	10 dB	22 dB	0 dB	18 dB
GA	−20 dB	18 dB	10 dB	22 dB	−24 dB	10 dB
Convex	−65 dB	18 dB	−10 dB	20 dB	−10 dB	19 dB

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
