# Peer review of "Beamforming Optimization in Internet of Things Applications Using Robust Swarm Algorithm in Conjunction with Connectable and Collaborative Sensors"

_sensors, 2020, doi:10.3390/s20072048_

Round 1
Reviewer 1 Report
The authorship propose a new algorithm for the synthesis of several geometries of Collaborative Beamforming (CB) of virtual sensor antenna arrays with maximum mainelobe and minimum sidelobe levels as well as null control using Canonical Swarm Optimization (CPSO) algorithm. It is a solid work.
Some Swarm Optimization algorithms published in reputation journals such as KBS should be reviewed in the related works.
Author Response
Dear Ms. Charlotte Li
Assistant Editor, MDPI Sensor Journal
Reference: sensors-731640
Authors: Mohammed Zaki Hasan, Hussain Al-Rizzo,
Paper Title: Beamforming Optimization in Internet of Things Applications Using Robust Swarm Algorithm in Conjunction with Connectable and Collaborative Sensors
Upon serious consideration of the Editor-in-Chief’s and the Reviewers’ comments of the above-mentioned manuscript, the authors have edited the submission accordingly and wish to have it re-reviewed for possible publication in MDPI Sensors (ISSN 1424-8220).
Before responding to the Reviewers’ comments, we would like to sincerely acknowledge their valuable comments and suggestions, which will ultimately result in a significant improvement to the original manuscript. In response to their comments we offer the following in Action Column in bold/italic:
Author's Reply to the Review Report (Reviewer 1)
The authorship propose a new algorithm for the synthesis of several geometries of Collaborative Beamforming (CB) of virtual sensor antenna arrays with maximum mainelobe and minimum sidelobe levels as well as null control using Canonical Swarm Optimization (CPSO) algorithm. It is a solid work.
Some Swarm Optimization algorithms published in reputation journals such as KBS should be reviewed in the related works.
Answer:
We have cited and evaluated the most appropriate/related literature from reputable journals including KBS. The following additional references have been added from KBS:
33.X. Cai, H. Qiu, L. Gao, C. Jiang, X. Shao, An efficient surrogate-assisted particle swarm optimization algorithm for high-dimensional expensive problems, Knowl.-Based Syst. Vol. 184, pp. 104901, 2019.
34.Liu, Y., et al., An affinity propagation clustering based particle swarm optimizer for dynamic optimization. 2020: p. 105711.
35.Zhang, Y., et al., Binary PSO with mutation operator for feature selection using decision tree applied to spam detection. 2014. 64: p. 22-31.
36.Xiang, Z., et al., An adaptive integral separated proportional-integral controller based strategy for particle swarm optimization. 2020: p. 105696.
- Yu, K., X. Wang, and Z.J.K.-B.S. Wang, Multiple learning particle swarm optimization with space transformation perturbation and its application in ethylene cracking furnace optimization. vol. 96, pp. 156-170, 2016.
Sincerely,
Dr. Mohammed Zaki Hasan
Professor Dr. Hussain Al-Rizzo

Reviewer 2 Report
In this paper, the beamforming optimization using CPSO has been proposed. However, I have some comments to improve the paper.
1) How do you consider the model of IoT?
2) Could you give the optimized beamforming? What is the difference between these different arrays? Please give the comparisons of the beam patterns.
3) The beam width for different method is so different. WHy? If the array is same, do you think the beamwidth is different?
4) Form Fog.8 and 9, the proposed menthod can be used practical applications. Please compare with the proposed method with more methods.
5) Do you consider the mutual coupling between the array elements?
6) In Fig.10, please clearify what is the new advantages of your proposed method.
7) How about the complexity of the proposed method?
8) If the array elements are distributed radomly, how do control the position of the elements?
9) The following antenna array optimizations should be included in the paper.
A novel block sparse reconstruction method for DOA estimation with unknown mutual coupling
Robust Beamforming Technique with Sidelobe Suppression Using Sparse Constraint on Beampattern
Null Broadening and Sidelobe Control Algorithm via Multi-Parametric Quadratic Programming for Robust Adaptive Beamforming
Controllable sparse antenna array for adaptive beamforming
An Improved Null Broadening Beamforming Method based on Covariance Matrix Reconstruction
Author Response
Dear Ms. Charlotte Li
Assistant Editor, MDPI Sensor Journal
Reference: sensors-731640
Authors: Mohammed Zaki Hasan, Hussain Al-Rizzo,
Paper Title: Beamforming Optimization in Internet of Things Applications Using Robust Swarm Algorithm in Conjunction with Connectable and Collaborative Sensors
Upon serious consideration of the Editor-in-Chief’s and the Reviewers’ comments of the above-mentioned manuscript, the authors have edited the submission accordingly and wish to have it re-reviewed for possible publication in MDPI Sensors (ISSN 1424-8220).
Before responding to the Reviewers’ comments, we would like to sincerely acknowledge their valuable comments and suggestions, which will ultimately result in a significant improvement to the original manuscript. In response to their comments we offer the following in Action Column in bold/italic:
Author's Reply to the Review Report (Reviewer 2)
- How do you consider the model of IoT?
Answer
There are several challenges that must be considered when a model is developed in the realm of virtual beamforming in IoT environments. For example, each sensor must connect with others as well as with the wireless devices involved in the IoT environment that may use different technologies for communicating with sensors. Generally, coexistence is a severe challenge in the IoT model, since the devices that are trying to communicate may interfere with each other resulting in data loss, rendering the sensors useless. Meanwhile, there is a substantial work on coexistence for collaborative beamforming; the forecast includes thousands of devices in a narrow range. These applications in IoT also impose challenging requirements on manufacturing as far as creating large-scale deployments is concerned. In the context of our paper, sensors within IoT environment are considered as an area of interest that acts as a collaboratively distributed antenna array that forms a specific network topology such as linear, mesh, ring, tree or even random according to the connectivity among the collaborative sensors. These collaborative sensors adjust the initial phases of their carriers such that signals from individual sensors add constructively and form a beam towards the direction of the intended base station and nulls towards interferers. Moreover, CB has the ability to increase the coverage area of WSN, and can also be viewed as an alternative scheme to the multihop relay communications. Consequently, CB brings many advantages compared to the multihop relay communications due to several reasons. Firstly, no dependency of communication quality on individual sensors. Secondly, CB enables operating a single-hop directly to the base-station, thus, it reduces the data overhead and delay.
- Could you give the optimized beamforming? What is the difference between these different arrays? Please give the comparisons of the beam patterns.
Answer
The directivity of the array in general is a function of the array geometry, spacing, among the elements, orientation and radiation pattern of the individual elements of the antenna array and the excitation (input signal) of each element that is found from the optimization model. The difference between the different array geometries is explained in the system model, page. We have provided the optimized beam patterns in Figures 7, 8, 9 and 10, and are also listed in Tables 3, 4, 5, and 6 using different optimization methods to validate the effectiveness of the proposed algorithm.
- The beam width for different method is so different. WHy? If the array is same, do you think the beamwidth is different?
Answer
The choice of optimization methodology depends on several limitations imposed by the sensor network such as the number of sensors deployed in a specific area and the application scenarios in IoT. Existing algorithms for node selection may cause some sensors in the network to be used more often than others and hence exhaust their batteries. If all sensors selected from an excessively large area in the network are utilized to generate a narrow mainlobe, then the lifetime of network might exhaust as well as the lifetime of frequently utilized sensors. The main costs for these algorithms are computing distances. In order to isolate the effect of randomization on time efficiency, we propose changing the process of randomization of swarm performance beyond convex optimization by selecting the further points i.e. solutions from an outside search space. Whether each selected solution swapped with revocable solution, then it could create many other new solutions for multimodal problems. Furthermore, the gain of mainbeam is independent of the boresight direction; it is always equal to the number of sensors N. As N increases, the mainbeam of each sensor can reach a greater distance from the neighbor, while its width stays the same. Collaborative beamforming concentrates the radiated power in a certain direction toward the base-station. We expect that the sensors near to the base-station have high probability of being connected together via their mainbeams. The size of the connectivity area is proportional to the gains of the mainbeam. Thus, the radiated power is fixed at each sensor, where is an ideal distributed beamforming with fixed number of collaborative sensors results in maximum gain whereas the expected value of power received at the base-station increases as N increases. On the other hand, the probability of two sensors facing each other in the random deployment is independent of the distance from the base-station. Therefore, the connectivity performance of random beamforming isalmost the same throughout the network. The gain of mainbeam is independent of theboresight direction; it is always equal to the number of sensors N. As N increases, the mainbeam of each sensor can reach a greater distance from the neighbor, while its width stays the same. Meanwhile, the randomness of positions and the number of sensors cause variations in the respective beampatterns, whereas both the sidelobes’ positions and magnitude are affected by the network topology. To ensure the connectivity among these collaborative sensors, the Cartesian product for each cluster grouped. Each sensor can randomly select its main beamforming direction and then progressively change its direction based on robustness of particle swarm optimization by acquisition of the pre-knowledge of locations of all its neighbors. In other words, starting with initial random weights, the selection of sensors can refine the weights based on sharing the information among them. Figure 5 shows the relative positions of collaborative sensors, where d, and f0, q0 denote the distance and angle between two connected sensors, respectively. All sensors are Gaussian distributed in (0, 2π]. To conclude the same array geometry possesses different gains and hence beamwidths depending on the complex weight or excitation signal applied to the antenna input terminals of each sensor node.
- Form Fog.8 and 9, the proposed method can be used practical applications. Please compare with the proposed method with more methods.
Answer
Results from the CPSO given in Figs. 8 and 9 depict almost similar beamforming patterns compared to the conventional beamforming patterns. Each sample of beamforming in UMA indicates at first a deterministic and then closely approximates the mainlobe of conventional beamforming pattern achieving the objective function better than the convex optimization and below the conventional beamforming pattern. However, this implies that the beamforming pattern of each sample does not spread away from the average of the conventional beamforming pattern. This is because the proposed selection algorithm based on CPSO acquires a new diversity of solution from outside the area of interest but inside its population solutions by sampling the solution instead of generating a new one from individual or several solutions. Furthermore, the updating velocity in Eq. 27 is adjusted based on sampling of solution to guide the future search toward the global solution and then to produce improved samples closer to the optimal solution. We have noticed that the differences in the mainlobe and sidelobe performance among the three algorithms are obvious within increasing the collaborative sensors in the antenna array. On the other hand, the gap of the mainlobe as seen in Figs. 9a and 9b becomes smaller in the three algorithms as it is illustrated in Figs. 9c, and 9d. CPSO shows a clear advantage of selecting collaborative sensors in the reduction of the sidelobe below 0dB which is essential in reducing interference and saving the lifetime of each sensor. From the results, it is noted that the CPSO overcomes the undesired increment of the sidelobe in conventional beamforming pattern and convex optimization. We believe that the connectivity solutions in the realm of IoT aim not only at supporting the needs imposed by the several practical applications (such as logistics, agriculture, and asset management), but also to support network self-management for these applications. Thus, a part of the utilization of the performance of CB is to improve the connectivity solutions for a certain application depends on how well its features solve the specific needs of the end application. There are several researches that address the comparison of connectivity solutions for IoT applications by covering many possible measurement domains with relevant parameters for topologies in terms of power or signal strength-based, time-based, and space-based.
- Do you consider the mutual coupling between the array elements?
Answer
Mutual coupling is usually taken into consideration if the sensor node or base station are equipped with multiple antennas. This is done by rigorously modeling the antenna array geometry (usually two to four elements per node or more in base stations) using commercial software packages such as CST or HFSS. These software tools are based on solving Maxell’s equations given the geometry, material used, boundary conditions and excitation of each element of the array (per a single sensor node or base station). However, in the problem under consideration, it is not possible to riorously model the “virtual” array due to the extremely large number of elements involved not to mention the random path loss which cannot be handled by existing tools. In all regular configurations considered, except the random array, we have spaced two adjacent sensor elements by a minimum of half free-space wavelength as this separation ensures the minimum accepted mutual coupling effects (mutual coupling effects are ignored at this separation). For the random array, however, we have used the traditional Array Factor approach which ignores mutual coupling but considers the spacing and related phase shift. We should note that this approach is considered in almost all literature related to random arrays in virtual beamforming.
- In Fig.10, please clearify what is the new advantages of your proposed method.
Answer
Results from the CPSO given in Figure 10 depict that CPSO forms a network topology with its objective function given in Eq. 29 constrained by Eqs. 30 evaluated by investigating the relationship between distance (i.e. the transmission range) and the connectivity obtained by sampling the selected collaborative sensors to achieve a beam towards a specific direction. This case applies to military applications whereas collaborative sensors are distributed randomly without supporting facilities (such as energy sources or wired) to specified position close to enemy territories. Consequently, their positions may be or not close to the area of interest and the transmission requires high directivity at the desired direction. Multiple sensors are deployed form multihop topology which leads to difficulty in regulating these sensors as a specific topology because of the environmental constraints such as the sensors’ position and connectivity with their neighbors. It can be concluded that the capability of CPSO in improving the performance of CB through satisfying the connectivity constraints is revealed when designing practical random antenna arrays in various IoT applications.
- How about the complexity of the proposed method?
Answer
The complexity of the proposed method depends on sensor locations which provides additional degree of freedom for controlling the sidelobes. In order to achieve the desired mainlobe accompanied by reduction of the sidelobes, it is required to select a subset of connected and collaborative sensors from candidate sensors within the same coverage area of each k-th source-destination pair. Therefore, the complexity is related to sampling that assumed to be as Latin distributed random variable which represents the whole coverage area considering the fluctuations/shadowing effects in the channel. Latin distribution depends on the distance among the selected collaborative sensors, whereas the sensors are close to each other, while the base-station is located far from these selected collaborative sensors. Moreover, the number of distributed collaborative sensors samples for achieving an optimal solution is an important factor, since each selected sensor must share the transmission range among others within the active cluster. This implies that the characteristic parameters of a CB of the selected sensors must maintain their stable values as the number of samples increases. Moreover, while testing one group or a group of sensors, we need to check if beamforming of the corresponding distributed collaborative sensors sample increases the mainlobe and decreases sidelobes in the intended and unintended direction(s), respectively. Thus, the minimum number of these samples can be determined when the mainlobe and sidelobes reach their stable values. We have explained this in pages 14-16, Sidelobe reduction via optimizing the sensor selection algorithm, paragraph subparagraph linear antenna array, lines 422- 443 and The Robust Canonical Particle swarm subparagraph, lines 361-384.
- If the array elements are distributed radomly, how do control the position of the elements?
Answer
Sensors within an area of interest act as collaboratively distributed antenna array that forms a specific network topology such as linear, mesh, ring, tree or even random according to the connectivity among the collaborative sensors. These collaborative sensors adjust the initial phases of their carriers such that signals from individual sensors add constructively and form a beam toward the direction of the intended base station. Moreover, CB has the ability to increase the coverage area of WSN, and can also be viewed as an alternative scheme to the multihop relay communications. Consequently, CB brings many advantages compared to the multihop relay communications due to several reasons. Firstly, no dependency of communication quality on individual sensors. Secondly, CB enables operating a single-hop directly to the base-station, thus, it reduces the data overhead and delay. Finally, CB in directional antenna transmission achieves higher connectivity compared omnidirectional transmission with the same transmit power.
The spacing among the collaborative sensors affects the beampattern shape of the antenna array. Thus, selecting random sensors positions i.e. uncertain sensor positions may lead to positions errors. However, the minimum SLL of an antenna array with fixed and selected sensors positions is higher than that of the antenna array with randomly selected sensors if the number of antenna array is the same, since the distance among the randomly selected sensors has maximum Euclidean distance corresponding to a certain hop-distance which is considered as a practical metric for modeling spatially random sensor network. Specifically, the connectivity in terms of estimating the area of interest is defined by maximum distance that can be covered in multi-hop paths. Furthermore, the maximum Euclidean distance is directly related to the estimated hop-distance which is equal to the least number of hops overall multi-hop paths between any two locations. Therefore, CPSO is used for the optimization of sensor selection whereas the topology of random selection becomes regular topology such as linear, mesh, tree, or ring and the SLL deteriorates further.
- The following antenna array optimizations should be included in the paper.
- A novel block sparse reconstruction method for DOA estimation with unknown mutual coupling
- Robust Beamforming Technique with Sidelobe Suppression Using Sparse Constraint on Beampattern
- Null Broadening and Sidelobe Control Algorithm via Multi-Parametric Quadratic Programming for Robust Adaptive Beamforming
- Controllable sparse antenna array for adaptive beamforming
- An Improved Null Broadening Beamforming Method based on Covariance Matrix Reconstruction
Answer
The five bullets list published journal articles. We have cited and evaluated the most appropriate references.
Sincerely,
Dr. Mohammed Zaki Hasan
Professor Dr. Hussain Al-Rizzo

Reviewer 3 Report
The authors propose the use of an optimized collaborative beamforming technique to improve the IoT nodes communication to the base station. The authors also propose optimizations to reduce the sidelobes levels.
Although I found a very interesting and promising approach, I think that the authors do not properly validate their results. Many problems are purely approximated through models, without the proper validation of their assumptions. For instance, the distribution of the multiple hops needs to be validated. Despite real experiments with certain amount of IoT nodes would be more convenient, network simulators such as Cooja can already provide more realistic node’s behavior.
There are also challenges and problems that are not covered, such as the additional communication cost between nodes. For instance, the weights are synchronized at each node. This demands additional communication between the nodes. I assume that such additional communication must happen only once, but how frequently should it happen? Each time a new node is connected to the network? There is a refresh period to discover new nodes? Moreover, it is also not clear for me who the synchronization problem is solved, which is in fact the main key for the constructive interference which amplifies the node’s communication to the base station.
The paper is mathematically well explained but it makes the paper dense and hard to follow in some parts. Perhaps I would suggest the authors to focus more in the beamforming main idea, validate it (not just mathematical models, but to use more realistic (and specialized) tools for IoT or to test with real WSN motes for instance. One can still find isolated areas where the wireless “pollution” is still low enough to perform such experiments. The proposed optimizations might be presented (and validated) once you have validated your models.
I would also recommend authors to review the paper. There are several grammar and spelling mistakes. For instance:
- This paper proposes a a local…
- Mainelobe instead of main lobe
- The sensor is able to communication…
- …
Despite my rejection due to the lack of validation, I would like to encourage the authors since I believe it is a promising approach.
Author Response
Dear Ms. Charlotte Li
Assistant Editor, MDPI Sensor Journal
Reference: sensors-731640
Authors: Mohammed Zaki Hasan, Hussain Al-Rizzo,
Paper Title: Beamforming Optimization in Internet of Things Applications Using Robust Swarm Algorithm in Conjunction with Connectable and Collaborative Sensors
Upon serious consideration of the Editor-in-Chief’s and the Reviewers’ comments of the above-mentioned manuscript, the authors have edited the submission accordingly and wish to have it re-reviewed for possible publication in MDPI Sensors (ISSN 1424-8220).
Before responding to the Reviewers’ comments, we would like to sincerely acknowledge their valuable comments and suggestions, which will ultimately result in a significant improvement to the original manuscript. In response to their comments we offer the following in Action Column in bold/italic:
Author's Reply to the Review Report (Reviewer 3)
The authors propose the use of an optimized collaborative beamforming technique to improve the IoT nodes communication to the base station. The authors also propose optimizations to reduce the sidelobes levels.
Although I found a very interesting and promising approach, I think that the authors do not properly validate their results. Many problems are purely approximated through models, without the proper validation of their assumptions. For instance, the distribution of the multiple hops needs to be validated. Despite real experiments with certain amount of IoT nodes would be more convenient, network simulators such as Cooja can already provide more realistic node’s behavior.
Answer
Many studies focused on designing MAC and routing protocols such as WisMAC and RPL for several IoT environments which are evaluated by simulation and it is unclear if implementations for real sensor hardware exist. Moreover, research on the complexity of some of these protocols and the absence of implementation that work on real sensor is still scarce and is outside the scope of our paper. Therefore, the evaluation and adaptation of any proposed protocol or approach is based on conventional approaches starting from theoretical formulation, simulations and finally experimental implementation, if possible. There are several challenges that must be considered in IoT environment in the evaluation and adaptation before the experimental executions, for example each sensor must connect with others as well as with the wireless devices that may use different technologies for communicating with sensors.
Generally, coexistence is a severe challenge in IoT model, since the devices that are trying to communicate may interfere with each other resulting in data lost, rendering the sensors useless. Meanwhile, there is substantial work on coexistence for CB, the forecast includes thousands of devices in a narrow range. This also considers the application of IoT requirements to manufacturing based on the possibility of creating large-scale deployments. Furthermore, the channel characteristics in various environments such as outdoor, or outdoor-to-indoor IoT applications, i.e. scenarios such as open-loop or closed-loop in which CB is needed are exposed to large-scale fading which is considered as a dominant factor for the channel among these collaborative sensors and base-station. Therefore, the channel coefficient for each k-th collaborative sensor which serves as connectable source-destination pair is multiplied by the corresponding weight in order to align the phase of the signal. This ensures that signals from all collaborative sensors are in-phase towards the direction of the intended base-station. Moreover, a closed loop scenario is considered where the phase alignment is done by compensating the distance between the collaborative sensors and the intended base-station, with respect to the cluster head. In all cases considered, a convergent test is always conducted to determine the size of elements/nodes/sensors in the problem under consideration. The implementation presented promising results whereas the efficiency is limited by the exhaustive search of linear network topology with all possible connectivities. This limitation is mainly due to the memory of particles i.e. the and . Changing the inertia cannot move the swarm out of the current optimum when velocity=0. Therefore, the convergent tests have been conducted when increasing the number of selected collaborative sensors in random deployment or even in a mesh deployment compared to linear deployment whereas the improvement decreases with increasing connectivity among the selected sensors. This resulted in an unacceptably long run times and time intensive data handling. One of the most impressive results in increasing population for a large number of convergence tests is to restart the number of sensors deployed in turn to find robust solutions and more reliably than the traditional particle swarm optimization algorithm.
There are also challenges and problems that are not covered, such as the additional communication cost between nodes. For instance, the weights are synchronized at each node. This demands additional communication between the nodes. I assume that such additional communication must happen only once, but how frequently should it happen? Each time a new node is connected to the network? There is a refresh period to discover new nodes? Moreover, it is also not clear for me who the synchronization problem is solved, which is in fact the main key for the constructive interference which amplifies the node’s communication to the base station.
The paper is mathematically well explained but it makes the paper dense and hard to follow in some parts. Perhaps I would suggest the authors to focus more in the beamforming main idea, validate it (not just mathematical models, but to use more realistic (and specialized) tools for IoT or to test with real WSN motes for instance. One can still find isolated areas where the wireless “pollution” is still low enough to perform such experiments. The proposed optimizations might be presented (and validated) once you have validated your models.
Answer
The deployment of sensors has been studied to explain the increasing intensity touching various design aspects and practical use cases. Several implementations have become available in common IoT operating systems focusing on the manufacturing stage of smart products lifecycle with the dynamic response to demand changes. Industrial safely and control system in most IoT applications are increasingly interconnected to exchange information and operational conditions of machines or equipment locally and report their status updates via sensors to external observers. The general communication requirement of IoT application is Collaborative Beamforming due to using low-cost, resource-constrained devices and network flexibility and scalability. CB requirements are specific to each application domain and vary in QoS parameters, availability, reliability and security. Practically, during the CB step, each collaborative node is first synchronized with the initial phase using the knowledge of the node locations. Alternatively, the synchronization can be performed without any knowledge of the node locations. For example, the synchronization algorithm of several studies uses a simple 1-bit feedback iterations, while others are based on the time-slotted round-trip carrier synchronization approach. Moreover, this comment brings issues that are valid but are outside the scope of our paper as the problem we are considering is tackling CB at the band-pass (radiation pattern) level rather than base-band and is independent of transmission standard/protocol. However, we are planning to extend our current work to include the MAC protocol by investigate the duty cycle of each sensor in the network by considering the probability of connectivity using Contiki cooja simulator. We have started to develop our proposed routing and MAC protocol using Arduino/Genuino Uno and ZigBee which is a microcontroller-board based on the ATmega328P to allow the users and designers to use this model and implements the discrete-event simulation to validate results. From this point of view, we are attempting to help the users and designers to build new specifications for node architectures and application profiles for different IoT operating systems. Our goal is to decide whether it makes sense to develop a new version of multimedia sensor node that supports the partitioning approach for detecting motion instead of periodically sampling of the camera.
I would also recommend authors to review the paper. There are several grammar and spelling mistakes. For instance:
- This paper proposes a a local…
- Mainelobe instead of main lobe
- The sensor is able to communication…
Answer
We have corrected grammar and spelling mistakes.
Sincerely,
Dr. Mohammed Zaki Hasan
Professor Dr. Hussain Al-Rizzo

Round 2
Reviewer 2 Report
It can be accepted.
This manuscript is a resubmission of an earlier submission. The following is a list of the peer review reports and author responses from that submission.
Round 1
Reviewer 1 Report
This paper proposed to use Canonical Swarm Optimization (CPSO) algorithm for Collaborative Beamforming (CB) of virtual sensor antenna arrays. The objectives include maximum mainelobe, minimum sidelobe and null control by using the node selection. The authors conducted MATLAB simulations to compare their proposed algorithm with convex optimization, Genetic Algorithm (GA), and Particle Swarm Optimization (PSO). However, the manuscript can not be acceptable in the current presentation. There are some essential issues should be improved:
1) The most important problem is that the manuscript is still very rough and lengthy, just writing with the roaming thought without considering the readers. The contribution is just a little by proposing CPSO rather than PSO used by other literatures to solve optimization. Thus the manuscript can be compressed to within just 20 pages, avoiding the time waste of readers and reviewers. Many long sentences are completely not acceptable for non-native readers, such as: “We establish the CPSO algorithm using directional antenna model [7] to analyze local connectivity i.e. the probability of isolation of the sensor, as well as the overall global connectivity which means evaluating the probability of connectivity of at least one connected path that identifies the network topology in terms of ring, tree, mesh, and random from the entire network.” There are almost 60 words in one sentence. And in page 13, the long sentence “It should be noted such that each cluster of the WSN consists of a large number of sensors, thus the mainlobe of the beampattern is unstable for different number of N^k of different subsets N^k, as long as the coverage area does dynamically change according to IoT environment (i.e. indoor, outdoor, or rural areas monitoring) as well as the requirements of the applications in that environment” is actually the same meaning as it followed sentence “If the coverage area does not change and the sensor network is partitioned into different clusters which consists of a sufficiently large number of sensors then the beampattern is considered stable.”
2) In Eq. (1), there should be d_{ij} = sqrt( (x_i – x_j)^2 + (y_i – y_j)^2 )
3) Eq. (25) is difficult to comprehensive because h(x) and f(x) emerge without explanation and without any relation to last section.
4) In the paper, the authors do not give their motivation why to use CPSO.
5) The authors claimed that they compared the proposed method with convex optimization. In fact, the optimization theory can be divided into convex and non-convex, so convex optimization is a very large field including many solving algorithms. The authors should point out which algorithm they used. In addition, because convex optimization has been investigated completely, when use convex, the actual optimal solution can be obtained which is better than heuristics algorithms, unless the convex problem is the simplified version of the original problem.
6) The node selection technique is formulated based on CPSO and the exhaustive search over all possible network connectivities of sensors in the network. The exhaustive search is very low efficiency and difficult to scale.
Author Response
Dear Mr. Kevin Chen
Assistant Editor-in-Chief, MDPI Sensor Journal
Reference: sensors-666941
Authors: Mohammed Zaki Hasan, Hussain Al-Rizzo,
Paper Title: Beamforming Optimization in Internet of Things Applications Using Robust Swarm Algorithm in Conjunction with Connectable and Collaborative Sensors
Upon serious consideration of the Editor-in-Chief’s and the Reviewers’ comments of the above-mentioned manuscript, the authors have edited the submission accordingly and wish to have it re-reviewed for possible publication in MDPI Sensors (ISSN 1424-8220).
Before responding to the Reviewers’ comments, we would like to sincerely acknowledge their valuable comments and suggestions, which will ultimately result in a significant improvement to the original manuscript. In response to their comments we offer the following in Action Column and (in bold/italic):
Acknowledgment
The authors would like to sincerely acknowledge the anonymous reviewers for their constructive comments and suggestions which have ultimately resulted in significant improvement of the content and quality of the revised manuscript.
Author's Reply to the Review Report (Reviewer 1)
The most important problem is that the manuscript is still very rough and lengthy, just writing with the roaming thought without considering the readers. The contribution is just a little by proposing CPSO rather than PSO used by other literatures to solve optimization.Answer: This is a new formulation, which is entirely different from traditional particle swarm optimization.
Thus, the manuscript can be compressed to within just 20 pages, avoiding the time waste of readers and reviewers.
Answer: The manuscript has been rewritten with some paragraphs/sentences being removed resulting in 25 pages. However, we believe that the analytical formulation is not a waste of time but rather essential for the non-expert readers.
Many long sentences are completely not acceptable for non-native readers, such as: “We establish the CPSO algorithm using directional antenna model [7] to analyze local connectivity i.e. the probability of isolation of the sensor, as well as the overall global connectivity which means evaluating the probability of connectivity of at least one connected path that identifies the network topology in terms of ring, tree, mesh, and random from the entire network.” There are almost 60 words in one sentence. And in page 13, the long sentence “It should be noted such that each cluster of the WSN consists of a large number of sensors, thus the mainlobe of the beampattern is unstable for different number of N^k of different subsets N^k, as long as the coverage area does dynamically change according to IoT environment (i.e. indoor, outdoor, or rural areas monitoring) as well as the requirements of the applications in that environment” is actually the same meaning as it followed sentence “If the coverage area does not change and the sensor network is partitioned into different clusters which consists of a sufficiently large number of sensors then the beampattern is considered stable.”
Answer: The two long sentences as well as others have been edited.
In Eq. (1), there should be d_{ij} = sqrt( (x_i – x_j)^2 + (y_i – y_j)^2 ).
Answer: corrected.
(25) is difficult to comprehensive because h(x) and f(x) emerge without explanation and without any relation to last section.Answer:
We have explained the relation between the objective function, denoted as f(x) and the Latin hypercube, denoted h(x) to obtain a random selection of collaborative sensors based on the conditional probability of sensors being connected for each cluster. Section 5, page 11 and page 12 provide the explanation.
In the paper, the authors do not give their motivation why to use CPSO.Answer:
We have given our motivation as to why we used the CPSO in Section 6, “Sidelobe Reduction via Optimizing the Sensor Selection Algorithm.” Subsection 6.1, Second paragraph, page 12. As well as in Section 7, “Performance Evaluation” line number 346-348, page 17.
The authors claimed that they compared the proposed method with convex optimization. In fact, the optimization theory can be divided into convex and non-convex, so convex optimization is a very large field including many solving algorithms. The authors should point out which algorithm they used. In addition, because convex optimization has been investigated completely, when use convex, the actual optimal solution can be obtained which is better than heuristics algorithms, unless the convex problem is the simplified version of the original problem.Answer:
The key difference between the two methods (convex and non-convex) is that in convex optimization there can be only one optimal solution, which is globally optimal, else or it can be proven that there is no feasible area or solution to the problem. Non-convex optimization may have multiple locally optimal solutions/points requiring much tuning to identify whether the problem has no solution or if the solution is global. Hence, the efficiency in terms of computational time, of the convex optimization problem is much better than non-convex. Therefore, we proposed a robust Canonical Swarm Optimization algorithm that employs a local neighbor search method (i.e. sharing information among the selected sensors) to define a Cartesian product. The objective of a local neighbor search is to obtain optimal bounds of network connectivity between collaborative sensors and the intended base-station. Hence, a heuristics algorithm is usually much easier to deal with in comparison to a convex problem that might takes a lot of time. Finally, it is not our intent to compare the two optimization schemes in depth since is outside the scope of this manuscript.
The node selection technique is formulated based on CPSO and the exhaustive search over all possible network connectivities of sensors in the network. The exhaustive search is very low efficiency and difficult to scale.
Answer:
We have mentioned the mechanism of exhaustive search over all possible network connectivities in term of various network topologies and we described the low efficiency search with increasing number of deployed sensors in Subsection 7.1. Linear antenna array third paragraph page 17, line number 370.
Reviewer 2 Report
This paper presents a Canonical Swarm Optimization algorithm for the synthesis of geometries for Collaborative Beamforming of virtual antenna arrays in Wireless Sensor Networks scenarios. The paper is generally well written, does a thorough review of previous works and it has a sound methodology. There are however, a couple of questions that this reviewer would like to ask, all of them are related to practical issues of system implementation:
1. The selected area (1200 m x 1200 m) for your simulated performance evaluation is relatively large. Could you please give one or two practical examples of application?
2. Depending on the provided application, for that area the number of sensors could be in practice be larger than 256. It would be interesting to see evaluation results for a larger number of nodes.
3. Your results provided from Tables 2 to 6 and from Figures 7 to 10 are not consistent in the reported number of nodes. Sometimes go up to 36, others 100 and in others 256. Is there any particular reason why not to all Tables and Figures report results of up to 256 nodes?
4. Could you relate your simulation setup and conditions to more practical issues, as to what transmission standard or technology (LoRAWAN, NBIoT, etc.) are your considerations applicable?
Author Response
Author's Reply to the Review Report (Reviewer 2)
The selected area (1200 m x 1200 m) for your simulated performance evaluation is relatively large. Could you please give one or two practical examples of application?Answer:
The selected area (1200 m x 1200 m) is sufficiently covered due to the high gain of the optimized virtual array that is sufficient to cover the area of interest. IoT applications such as in military, industrial, or smart-city are considered to illustrate the benefits of the proposed CPSO algorithm As mentioned in manuscript.
Depending on the provided application, for that area the number of sensors could be in practice be larger than 256. It would be interesting to see evaluation results for a larger number of nodes.Answer:
We observed that a converged solution is sought and hence it is always conducted to determine the size of elements/nodes/sensors in the problem space. Therefore, the converged solution has been verified when increasing the number of selected collaborative sensors in random deployment or even in a mesh deployment against linear deployment, which in turn resulted in unacceptably long run times and time intensive data handling. One of the most impressive results in increasing population for a large number of converged solutions with controlled restarts the number of deployment sensors in concert to find robust solutions and more reliable than traditional particle swarm optimization algorithms. Finally, the gain obtained using the number of sensors considered is far beyond what is required to cover the areas considered.
Your results provided from Tables 2 to 6 and from Figures 7 to 10 are not consistent in the reported number of nodes. Sometimes go up to 36, others 100 and in others 256. Is there any particular reason why not to all Tables and Figures report results of up to 256 nodes?Answer:
The simulation scenarios have been revised and are consistent.
Could you relate your simulation setup and conditions to more practical issues, as to what transmission standard or technology (LoRAWAN, NBIoT, etc.) are your considerations applicable?Answer:
This is outside the scope of our paper as the problem under consideration is tackling CB at the radiation pattern level (passband/RF level) and is not dependent on transmission standard. However, we are in the process of extending our current work to include the MAC protocol by investigating the duty cycle of each sensor in the network by considering the probability of connectivity using Contiki cooja simulator.

Round 2
Reviewer 1 Report
The authors has rewritten long sentences and shrunk the paper. However, the authors have not deep insight to CPSO and convex optimization and there are some important issues that not be addressed yet:
The references using PSO to optimize beamforming are not cited and evaluated.
The authors have not idea about the difference between CPSO and PSO.
Although the authors have cited the reference “Antenna array pattern synthesis via convex optimization” and given the paper as supplement in the revision. But it seems that the authors just read the title so they have no idea about the classical inter-point method adopted by the reference. Thus I think they do not implement the convex optimization algorithm really.
Because the CPSO still searches the neighboring optimal point so the performance can not beyond convex optimization.
In the reply to our last comment 6, the authors does not address the low efficiency of exhaustive search over all possible network connectivities.
Author Response
Author's Reply to the Review Report (Reviewer 1)
The reference using PSO to optimize beamforming are not cited and evaluated
Answer
We have cited and evaluated the most appropriate reference that uses PSO. Some references have been removed as per your request to reduce the size of the paper.
The authors have not idea about the difference between the CPSO and PSO
Answer
The Particle Swarm Optimization (PSO) algorithm is inspired by social behavior patterns of organisms that live and interact within large groups. It could be easily implemented and applied to solve various optimization problems or problems that can be transformed to function optimization. PSO is main strength is fast convergence, which compares favorably with many other global optimization algorithms such as canonical particle swarm optimization (CPSO), fully-informed particle swarm optimization, and multi-swarm optimization algorithms. The exploration and exploitation trade-off is improved for general swarms to include the influence of the inertia term.
CPSO provides a formal proof that each particle converges to a stable point using stochastic process theory. Therefore, CPSO has the ability to analyze the stochastic convergence of the particle swarm and corresponding parameter selection guidelines. It has been shown
that the trajectories of the particles oscillate as different sinusoidal waves and converge quickly. Various methods have been used to identify other particles that influence the individual. For more information there are several books that discuss the differences between PSO and CPSO that we would like the reviewer to refer to that we are very much aware of:
1. Christian, Blum, and Merkle Daniel. "Swarm Intelligence Introduction and Application." Natural Computing Series. Springer, 2008.
2. Yang, Xin-She, et al., eds. “Swarm intelligence and bio-inspired computation: theory and applications”. Newnes, 2013.
3. Clerc, Maurice. “Particle swarm optimization”. Vol. 93. John Wiley & Sons, 2010. Chapters 7, 8,9,10, and 11, pages 87-144.
4. Bansal, Jagdish Chand, Pramod Kumar Singh, and Nikhil R. Pal, eds. “Evolutionary and swarm intelligence algorithms”. Springer, 2019.
5. Engelbrecht, Andries P. Computational intelligence: an introduction. John Wiley & Sons, 2007. Part IV COMPUTATIONAL SWARM INTELLIGENCE, Chapter 16 page no. 285-370.
Indeed, this comment is inappropriate and is insulting. We have a very clear idea about the difference between CPSO and PSO; please also refer to our extensive record of publications on topics related to PSO and CPSO.
1. Al‐Turjman, Fadi, Mohammed Zaki Hasan, and Hussain Al‐Rizzo. "Task scheduling in cloud‐based survivability applications using swarm optimization in IoT." Transactions on Emerging Telecommunications Technologies 30.8 (2019): e3539.
2. Hasan, Mohammed Zaki, and Hussain Al-Rizzo. "Optimization of Sensor Deployment for Industrial Internet of Things Using a Multiswarm Algorithm." IEEE Internet of Things Journal 6.6 (2019): 10344-10362.
3. Hasan, Mohammed Zaki, and Fadi Al-Turjman. "SWARM-based data delivery in social Internet of Things." Smart things and femtocells. CRC Press, 2018. 179-218.
4. Hasan, Mohammed Zaki, and Hussain Al‐Rizzo. "Task scheduling in Internet of Things cloud environment using a robust particle swarm optimization." Concurrency and Computation: Practice and Experience 32.2 (2020): e5442.
5. Hasan, Mohammed Zaki, and Fadi Al-Turjman. "Optimizing multipath routing with guaranteed fault tolerance in Internet of Things." IEEE Sensors Journal 17.19 (2017): 6463-6473.
Although the authors have cited the reference “Antenna array pattern synthesis via convex optimization” and given the paper as supplement in the revision. But it seems that the authors just read the title so they have no idea about the classical inter-point method
adopted by the reference. Thus I think they do not implement the convex optimization algorithm really.
This comment has nothing to do with the manuscript. You are assuming that “they do not implement the convex optimization algorithm really” and your assumption is wrong.
In the reply to our last comment 6, the authors does not address the low efficiency of exhaustive search over all possible network connectivities.
Answer
We have read the cited reference “Antenna array pattern synthesis via convex optimization” and concluded that in the convex optimization algorithm, every local minimizer of a convex objective function over a convex set is a global minimizer, and hence needs gradient information. Therefore, similar to most of the traditional optimizers, the algorithm is unfit for complex multimodal problems and non-differentiable optimization problems.
Problems that arise in communication engineering applications have a number of other factors that influence the optimization process. Most notably, these factors are uncertainties and noise. For example, in various IoT applications, it may be very difficult to control the parameters with an arbitrary precision. Another example is when the system uses sensors to determine the quality of the solution. These sensors are likely to be limited in their precision and produce therefore inaccurate results.
These problems can be modeled as a linear convex optimization problem in order to obtain optimal and accurate solution. Moreover, it is not easy to detect whether a function is convex or not. Thus, the richness of convex functions for solving these problems is demonstrated by the connection between the convexity set and the objective function from classical analysis. This connection is given by several theorems to guarantee that the convexity is the main step in the optimization method.
In “Antenna array pattern synthesis via convex optimization”, the authors presented a solution for the single look direction of the antenna array problem using an interior-point method to define the connections with specific geometry. All large-scale interior-point implementations use direct decomposition to solve the reduced Newton system. This can be done by the symmetric factorization of a quasidefinite system.
Once the problem is defined as a convex optimization problem, it can be solved in a relatively short time using well-kown efficient solvers. However, one drawback of direct factorization approaches is that in some situations a sufficient/desired numerical accuracy cannot be achieved. Therefore, we utilized the CVX solver of the Matlab-based modeling system for comparing the efficient and robust implementation of a numerical algorithm with Canonical PSO by generating results for the convex optimization. CVX turns Matlab
into a modeling language, allowing constraints and objective functions to be specified using standard Matlab expression syntax.
In contrast, the CPSO considers only robustness against input parameters to model the problem where the input parameters are distributed according to a probability density function. It is impossible to compute the objective functions in Equation 20 analytically for the complex problem under consideration. Therefore, we used the Latin distributed random variable integration by sampling over a number of instances of the input parameters. This method has a drawback in that, as each sample corresponds to an objective function evaluation, the number of objective function evaluations increases. In the initial design and verification of a collaborative beamforming approach, the computational complexity is not a significant factor as compared to the accuracy of the results. Implementation of the algorithm for a specific real-life scenario is outside the scope of this paper.
On another hand, CPSO moves the particle by attractive force from the global best position (Gbest) and the particle’s own personal best position (Pbest) of the global solution. This mechanism can achieve a higher convergence rate. However, the CPSO suffers from premature convergence on complex multimodal problems. In the initialization phase of optimization, the diversity of canonical PSO is lost quickly and the algorithm’s exploration performance weakens compared with convex optimization. In the latter phase, the particles crowd in the neighborhood optimal solution of Gbest and slowly down the algorithm’s convergence rate.
However, to enhance the exploration of PSO, many improved PSO variants have been proposed. These PSO variants can be classified into five categories:
1. Adjustment of the configuration parameters to balance the global and local search abilities. This call is parameter tuning.
2. Enhancing the population diversity by designing new information propagation strategies. This is called neighborhood topology, where the proposed algorithm is evaluated by reinitializing the resampling of the search space according to different topology structures, which can be reutilized in CPSO, with different strategies to share search information for every particle. Mesh, global star and local ring are the three most commonly used structures. CPSO with mesh and global star structure, where all particles are connected to each other, have the smallest average distance and fastest propagation in all topologies in swarm. In the contrast, a CPSO with local ring structure, where every particle is connected to two near particles, has the largest average distance in swarm. The most important factor affecting an optimization algorithm’s performance is its ability of “exploration” and “exploitation”.
A better optimization algorithm should optimally balance the two conflicting objectives, in which the ability of exploration and ability of exploitation should be adjusted via the population diversity analysis when solving different problems or on different search stages. For example, to solve multimodal problem, great exploration ability means that the algorithm has great possibility to “jump out” of local optimal.
3. Hybridization of PSO algorithm and other auxiliary search techniques.
4. Introduction of multiple swarms or coevolving groups to improve the global search ability.
5. Combine PSO algorithms with new efficient learning strategies.
